# The transcriptomic landscape of monosomy X (45,X) during early human fetal and placental development
Jenifer P. Suntharalingham [1], Ignacio del Valle [1], Federica Buonocore [1], Sinead M. McGlacken-Byrne [1], Tony Brooks[2], Olumide K. Ogunbiyi[3,4,5], Danielle Liptrot[4], Nathan Dunton[2], Gaganjit K. Madhan[2], Kate Metcalfe[4], Lydia Nel[4], Abigail R. Marshall[4], Miho Ishida[1], Neil J. Sebire[3,4,5], Gudrun E. Moore [1], Berta Crespo [4], Nita Solanky[4], Gerard S. Conway [6] & John C. Achermann [1] ✉

Monosomy X (45,X) is associated with Turner syndrome and pregnancy loss in humans, but the underlying mechanisms remain unclear. We therefore undertook an exploratory study of the transcriptomic landscape of clinically relevant human fetal 45,X tissues (including pancreas, liver, kidney, skin, placenta) with matched 46,XX and 46,XY control samples between 11 and 15 weeks post conception ($n = 78$). Although most pseudoautosomal region 1 (PAR1) genes are lower in monosomy X tissues, we also found reduced expression of several key genes escaping X inactivation (e.g., *KDM5C* and *KDM6A*), several ancestral X-Y gene pairs, and potentially clinically important transcripts such as genes implicated in ascending aortic aneurysm. In contrast, *higher* expression of an autosomal, long non-coding RNA (*OVCH1-AS1*) is seen in all 45,X tissues. In the placenta, lower expression of *CSF2RA* is demonstrated, likely contributing to immune dysregulation. Taken together, these findings provide insights into the biological consequences of a single X chromosome during early human development and potential insights in genetic mechanisms in Turner syndrome.

Complete or partial loss of the second X (sex) chromosome in humans occurs in approximately 1:2500 girls and women and is associated with Turner syndrome (TS)[1,2]. Around 50% of individuals with TS have a monosomy X (45,X) karyotype but other variations in karyotype are often seen (e.g., isochromosome Xq (46,X,i(Xq)), ring X (mosaic 45,X/46,X,r(X)), Xp or Xq deletions, and 45,X/46,XX or 45,X/46,XY mosaicism)[1–4]. Although monosomy X is the only chromosome monosomy compatible with survival in humans, it is estimated that only a proportion of monosomy X fetuses survive to term and many pregnancies are spontaneously lost in the first or early second trimester, often before a pregnancy is recognized[5–7].

Girls and young women with TS/monosomy X can present with many different features and at different ages. In the newborn period, the diagnosis may be suspected due to lymphedema, congenital cardiovascular anomalies (e.g., coarctation of the aorta), renal features (e.g.,

horseshoe kidney) or distinct physical signs (e.g., wide chest, widened neck skin)[8]. Transient hyperinsulinism and hypoglycemia have also been reported[9,10]. In childhood, early features include impaired growth and recurrent otitis media, whereas absent puberty and primary ovarian insufficiency (POI) present later in teenage years[8,11,12]. Higher risks of long-term co-morbidities in adulthood are described, such as diabetes mellitus, weight gain, hypertension, raised liver enzymes, hearing impairment, hypothyroidism, autoimmunity, acquired cardiovascular disease, and skin nevi[12–16]. Women with TS have an overall increased mortality partly accounted for by aortic root dilatation and dissection[13,16–18]. Thus, the clinical features associated with TS can affect many different systems and may have some origins in early fetal development[2]. Identifying underlying mechanisms that drive the clinical features of TS is important as they may help to develop personalized medicine strategies and improve care for girls and women in the long term[1,2,19].

[1]Genetics & Genomic Medicine Research and Teaching Department, UCL Great Ormond Street Institute of Child Health, University College London, London, WC1N 1EH, UK. [2]UCL Genomics, UCL Zayed Centre for Research into Rare Disease in Children, UCL Great Ormond Street Institute of Child Health, University College London, London, WC1N 1DZ, UK. [3]Department of Histopathology, Great Ormond Street Hospital for Children NHS Foundation Trust, London, WC1N 3JH, UK. [4]Developmental Biology and Cancer Research and Teaching Department, UCL Great Ormond Street Institute of Child Health, University College London, London, WC1N 1EH, UK. [5]NIHR Great Ormond Street Biomedical Research Centre, London, WC1N 1EH, UK. [6]Institute for Women's Health, University College London, London, WC1E 6AU, UK. ✉e-mail: j.achermann@ucl.ac.uk

Key genetic mechanisms hypothesized to drive TS phenotypes are usually related to the complete or partial loss of the second sex chromosome[1,2,20,21]. Haploinsufficiency of genes in the pseudoautosomal (PAR) regions (PAR1, PAR2), which have Y chromosome homologs, have been associated with TS phenotypes such as short stature and cubitus valgus (e.g., *SHOX*) or cardiac QTc interval duration[2,22–24]. Monosomy X may also be associated with reduced dosage of genes that normally escape X-inactivation[2,20,21,25,26]. X inactivation is a process whereby one X chromosome is transcriptionally silenced in female mammalian cells to equalize dosage of gene products from the X chromosome between 46,XX and 46,XY individuals[7,25,27]. This process is mediated largely by the non-coding RNA transcripts, *XIST/TSIX*[25,27]. It is well established that some genes on the X chromosome escape X inactivation; these genes are normally biallelically expressed from both X chromosomes in 46,XX females, but this may not occur in those with a 45,X karyotype[20,21,25,28,29]. Moreover, a core subset of X chromosome genes that might drive somatic sex differences has recently been proposed[27].

In addition to direct effects of reduced PAR gene dosage, other mechanisms linked to the pathogenesis of TS include disruption of X chromosome genes that have a knock-on (or "ripple") effect on other parts of the X chromosome itself[2,20,21], including important non-coding RNAs such as *JPX* (a key regulator of the X inactivation gene, *XIST*)[20,26,30], or X chromosome genes that influence autosomal genes with regulatory functions such as ubiquitination, chromatin modification, translation, splicing, DNA methylation and circular RNA generation[2,20,21,26,28–32]. Unravelling these complex interactions requires whole genome transcriptomic analysis at a suitable scale and in relevant tissues.

To date, most transcriptomic studies investigating the pathogenic basis of TS have analyzed blood leukocytes/peripheral blood mononuclear cells (PBMCs)[21,26,29]. Direct sampling of tissues strongly associated with phenotypic features such as diabetes, hypertension and obesity is much more challenging. Recently, transcriptomic analysis of adult fat and muscle biopsies has been reported from individuals with TS and related sex chromosome aneuploidies such as 47,XXY (Klinefelter syndrome)[20,32]. This approach is starting to provide insight into the effects of sex chromosomes in different tissues, and to identify core haploinsufficient X chromosome genes associated with a 45,X karyotype[20,33,34]. However, the biochemical and transcriptomic profile of adult tissues may also be influenced by confounding factors such as medication, inflammation, diet or the complex interplay between different systems (e.g., fat, muscle and pancreas in insulin sensitivity), so assessing the "pure" monosomy X transcriptome is difficult.

Given the fact that many clinical features associated with monosomy X are present in early postnatal life, and that the origins of many long-term adult conditions may be in part established during embryonic or fetal development[35,36], we undertook detailed transcriptomic analysis of monosomy X fetal samples from several key tissues of interest between 11-15 weeks post conception (wpc). Although we were limited by sample availability to some extent, our aim was to better understand the transcriptomic events associated with monosomy X in early human development, and to obtain a unique perspective on human X chromosome biology and possible disease mechanisms in Turner syndrome, and to investigate the effects of monosomy X in the placenta in relation to pregnancy loss.

## Results
### Global transcriptomic differences and key sex chromosome genes
In order to identify global transcriptomic differences associated with a single X chromosome during development, we performed bulk RNA sequencing (bulk RNA-seq) using human monosomy X fetal samples (n = 20) between 11 and 15 wpc and compared these to tissue- and age-matched 46,XX (n = 20) and 46,XY controls (n = 20) (Fig. 1a, Supplementary Fig. 1 and Supplementary Data 1). Pancreas, liver, kidney, skin, and a mixed sample group (comprised of brain, heart, lung and spleen) were chosen, as these are biologically relevant to the clinical features associated with TS in childhood or in later life. All 60 tissue samples underwent single nucleotide polymorphism (SNP) array analysis on simultaneously extracted DNA. This approach confirmed the expected karyotype in all cases, except for one 45,X fetus where low-level mosaicism for a 46,XY cell line was seen (Supplementary Fig. 2).

Using principal component analysis (PCA) for the entire dataset (n = 60), samples clustered together based primarily on tissue of origin rather than karyotype, as expected (Fig. 1b, Supplementary Fig. 3 a, b). In contrast, PCA of individual tissues showed an influence of karyotype on clustering for pancreas (Fig. 1c) and liver (Fig. 1d), some effect of karyotype in kidney, and limited effect in skin (Fig.1 e, f and Supplementary Fig. 3 c, d).

Heatmap normalized expression of several important sex chromosome genes is shown in Fig. 1g. *XIST*, the principle X chromosome regulator of X inactivation, was strongly expressed in all 46,XX tissues, consistent with the presence of two chromosomes. *XIST* was not expressed in 46,XY samples nor in 45,X samples, both of which have a single X chromosome and do not undergo X inactivation. Y chromosome genes (e.g., *DDX3Y*, *KDM5D*, *RPS4Y1*, *USP9Y*) were strongly expressed in 46,XY samples and not in 46,XX samples nor 45,X samples, except at low level in the two mosaic samples (pancreas, heart/mixed) outlined above (Fig. 1g).

In order to identify genes with consistently lower or higher expression in monosomy X tissues compared to controls, differential gene expression (DGE) analysis was undertaken. Volcano plots comparing 45,X samples (n = 4) with either 46,XX or 46,XY matched control samples (n = 4 each group) for each tissue are shown in Fig. 2 (Supplementary Data 2–25). Genes with higher expression in 45,X samples have a positive log$_2$ fold change (log$_2$FC), whereas genes with a higher expression in either 46,XX or 46,XY samples have a negative log$_2$FC. As expected, the X inactivation genes *XIST* and *TSIX* showed higher expression in 46,XX samples compared to 45,X samples, whereas Y chromosome genes were differentially expressed in 46,XY samples compared to 45,X (Fig. 2).

### Genes with lower expression in monosomy X tissues
Our initial analysis focused on genes that were lower in both 45,X *versus* 46,XX and 45,X *versus* 46,XY datasets, and consistent across each of the five different tissues studied (n = 10 datasets) (log$_2$FC < -0.5, adjusted *p*-value (p-adj) < 0.05; Supplementary Fig. 4). This relatively low threshold for log$_2$FC was chosen to identify subtle but potentially meaningful differences in gene dosage, especially related to haploinsufficiency effects in monosomy X, and within the context of small sample sizes. By intersecting these groups, no genes were shared among all tissues studied with this cut-off, although four genes (*SLC25A6*, *AKAP17A*, *GTPBP6*, *ZBED1*) did have lower monosomy X expression in multiple tissues (Fig. 3a–c and Supplementary Fig. 4). These genes showed haploinsufficiency of expression in monosomy X tissues in violin plots (Fig. 3b and Supplementary Fig. 5). As these genes are all located in the PAR1 region of the X chromosome (Fig. 3d), investigation of the expression of all PAR genes was undertaken for 45,X *versus* 46,XX or 46,XY tissues (Fig. 3e). This analysis showed consistently lower expression of many PAR1 genes in 45,X tissues compared to controls, where log$_2$FC = -1.0 represents haploinsufficiency in 45,X, and where log$_2$FC = 0 represents similar expression in control and 45,X samples (Fig. 3e). Several PAR1 genes did not show differential expression; these genes generally had low transcript counts, or were not detected in all tissues (e.g., *SHOX*, *CRLF2*, *P2RY8*, *ASMT*, *XG*) (Supplementary Fig. 5 and Supplementary Data 26 and 28). Of note, the three PAR2 region genes (*SPRY3*, *VAMP7*, *IL9R*) also did not show marked differential expression (Fig.3e, Supplementary Fig. 6, and Supplementary Data 27 and 28), as shown in samples from adults with monosomy X[20,21,32].

In addition to PAR genes, recent reports have identified core sets of genes that likely escape X inactivation or influence phenotype in monosomy X adult tissues[20], as well as ten X chromosome genes that are implicated as key mediators of biological sex differences[27]. Analysis of these genes in monosomy X tissues during development compared to controls is shown in Fig. 4a–c, and Supplementary Data 29, 30. Notably, the histone demethylase genes, *KDM5C* (also known as *JARID1C/SMCX*) and *KDM6A* (*UTX*) had lower expression in 45,X compared to 46,XX control tissues, likely due to X

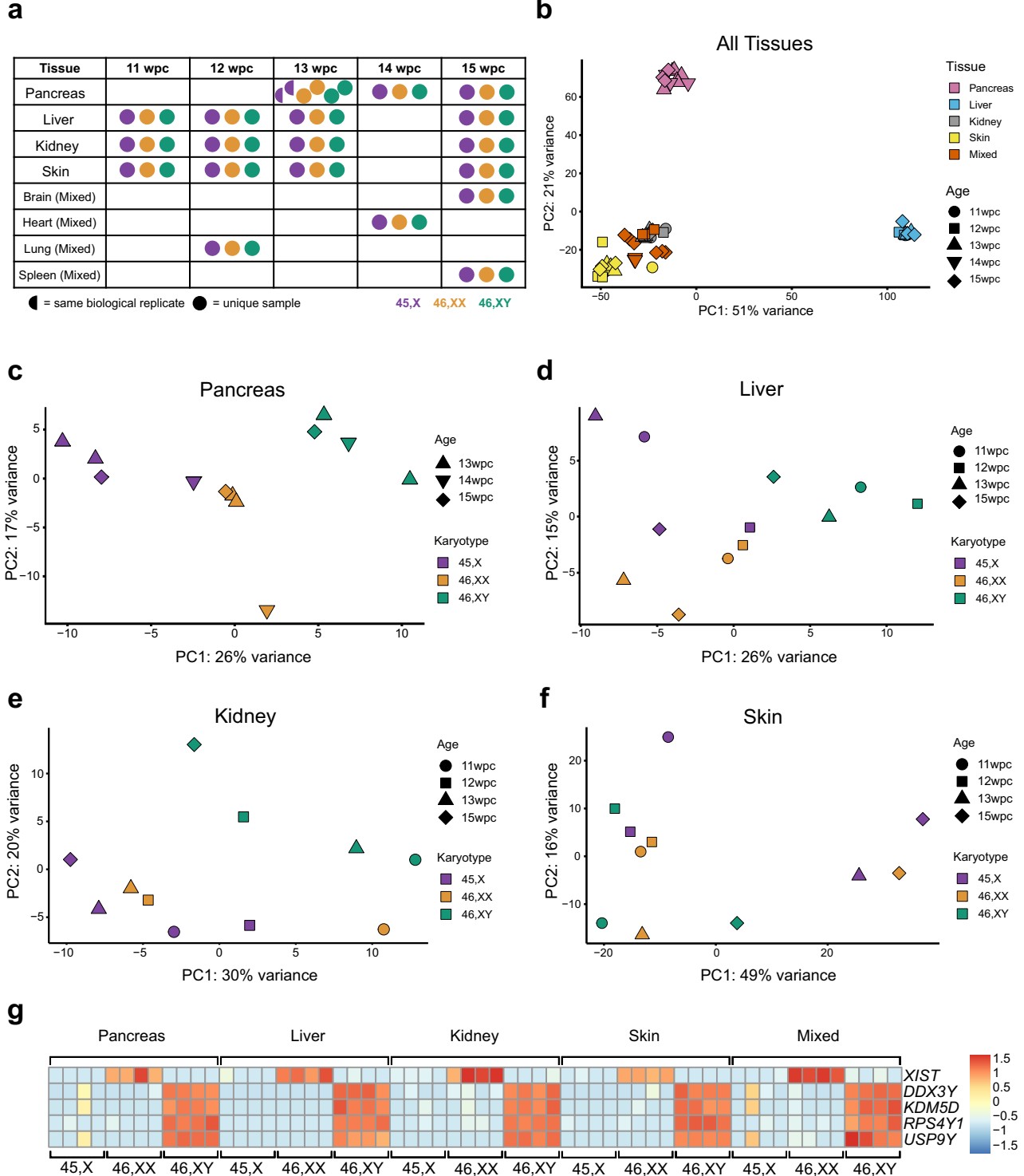

**Fig. 1 | Experimental design and principal component analysis of tissues.**
**a** Overview of age stages and tissues used for the main study. Karyotypes are indicated by the key. One 13 week post conception (wpc) pancreas was bisected longitudinally and both parts processed independently. **b** Principal component analysis (PCA) of the whole multi-tissue dataset ($n = 60$). PC, principal component. **c** PCA of pancreas samples ($n = 12$) (Note: the two 45,X samples at 13 wpc showed biological variability). **d** PCA of liver samples ($n = 12$). **e** PCA of kidney samples ($n = 11$; one outlier removed due to low level ( < 10%) adrenal contamination). **f** PCA of skin samples ($n = 11$; one outlier removed due to low level ( < 10%) muscle contamination). Age and karyotypes of tissues are shown in the key. **g** Heat map of key sex chromosome genes for each set of tissue samples; *XIST* for the X chromosome and *DDX3Y, KDM5D, RPS4Y1* and *USP9Y* for the Y chromosome. Expression intensity is shown in the key.

inactivation escape. As *KDM5C* and *KDM6A* are X-chromosome genes with Y-chromosome homologs (*KDM5D* and *UTY*, respectively), we extended our analysis to look at differential expression of homologous, ancestral X-Y gene pairs (as defined by Godfrey et al. 2020)[37] (Fig. 4d, e and Supplementary Data 31–33). Most X-Y gene pairs showed a decrease in both

45,X *versus* 46,XX and in 45,X *versus* 46,XY datasets, with the most consistent and marked lower expression in monosomy X for *EIF1AX/EIF1AY*, *ZFX/ZFY*, *DDX3X/DDX3Y*, *KDM6A/UTY* (also known as *KDM6C*), *KDM5C/KDM5D* (also known as *JARID1D/SMCY*), and *RSP4X/RSP4Y1*. Many of these genes play a role in transcription, translation and histone

**Fig. 2 | Volcano plots showing differential expression of genes between the 45,X and either 46,XX or 46,XY matched control samples.** Data are shown for: (**a**) Pancreas; (**b**) Liver; (**c**) Kidney; (**d**) Skin; (**e**) Mixed group (brain, heart, lung, spleen). Four samples were included in each group (see experimental design Fig. 1a). Comparison of 45,X versus 46,XX is shown in the left-hand panel and comparison of 45,X versus 46,XY is shown on the right-hand panel. The top ten most differentially expressed genes in each dataset are labeled, based on adjusted p-value and where $\log_2$ fold change (FC) is greater than +/−0.5. Genes with higher expression in 45,X tissues have a positive $\log_2$FC and those with higher expression in control samples have a negative $\log_2$FC. The significance level of highlighted points is shown in the key.

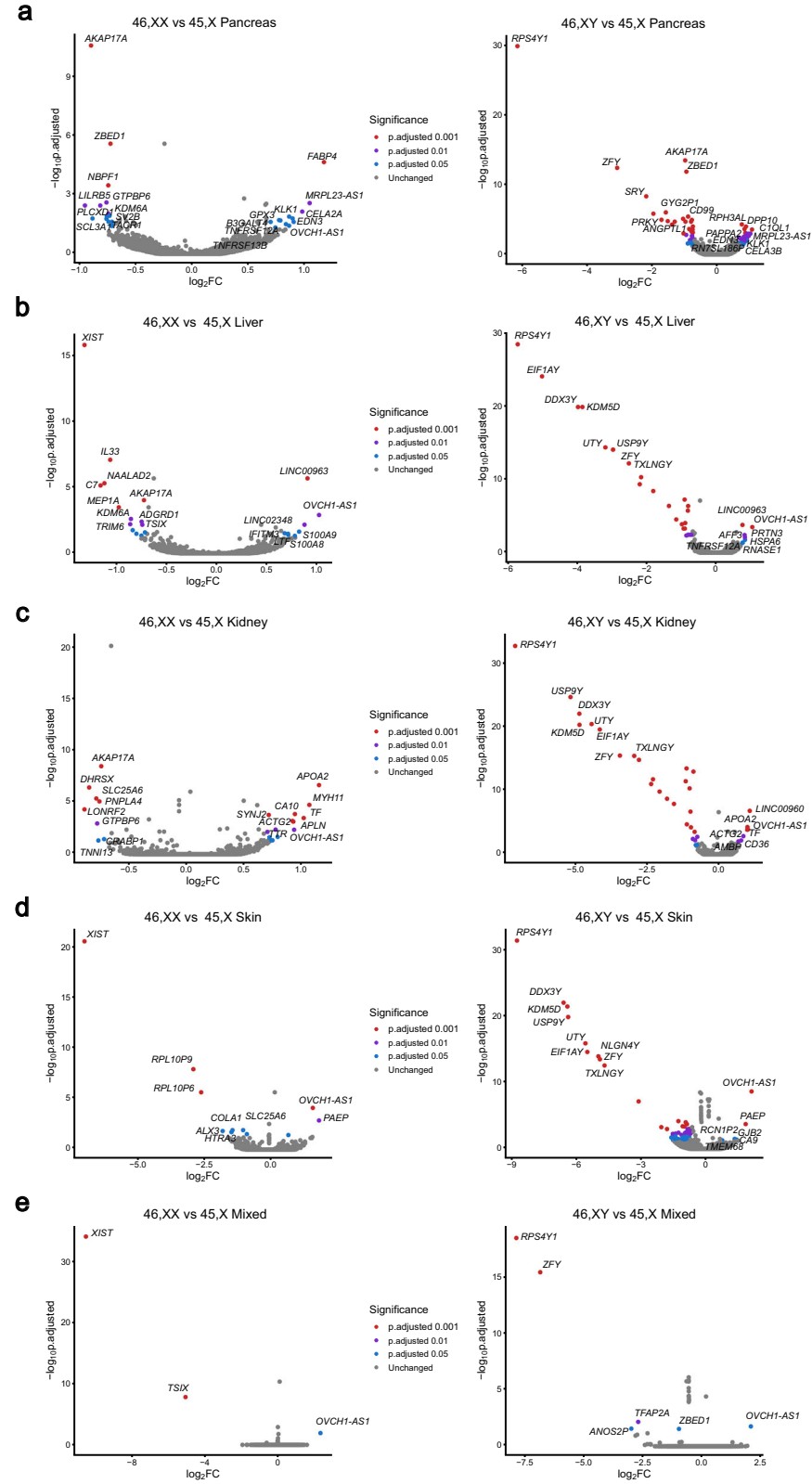

methylation status, which may be altered in 45,X and could influence developmental processes.

Next, we analyzed genes that showed global lower expression in monosomy X across multiple tissues (total $n = 10$; of which $n = 5$ were 45,X < 46,XX and $n = 5$ were 45,X < 46,XY). We hypothesized that these genes could also have important biological functions, beyond the PAR/XCI

model. In addition to the PAR genes and XCI escape genes described above, several notable autosomal genes (e.g., *ALDH1A3*, *RELN*, *LDLR*, *DKK2*) and other X chromosome genes (e.g., *AGTR2*) emerged as having lower expression in 45,X tissues (mean $\log_2$FC below −0.4 for 45,X compared to controls shown in Fig. 5a and Supplementary Data 34). Furthermore, when a $\log_2$FC below −0.35 cut off was applied to identify key genes for disease

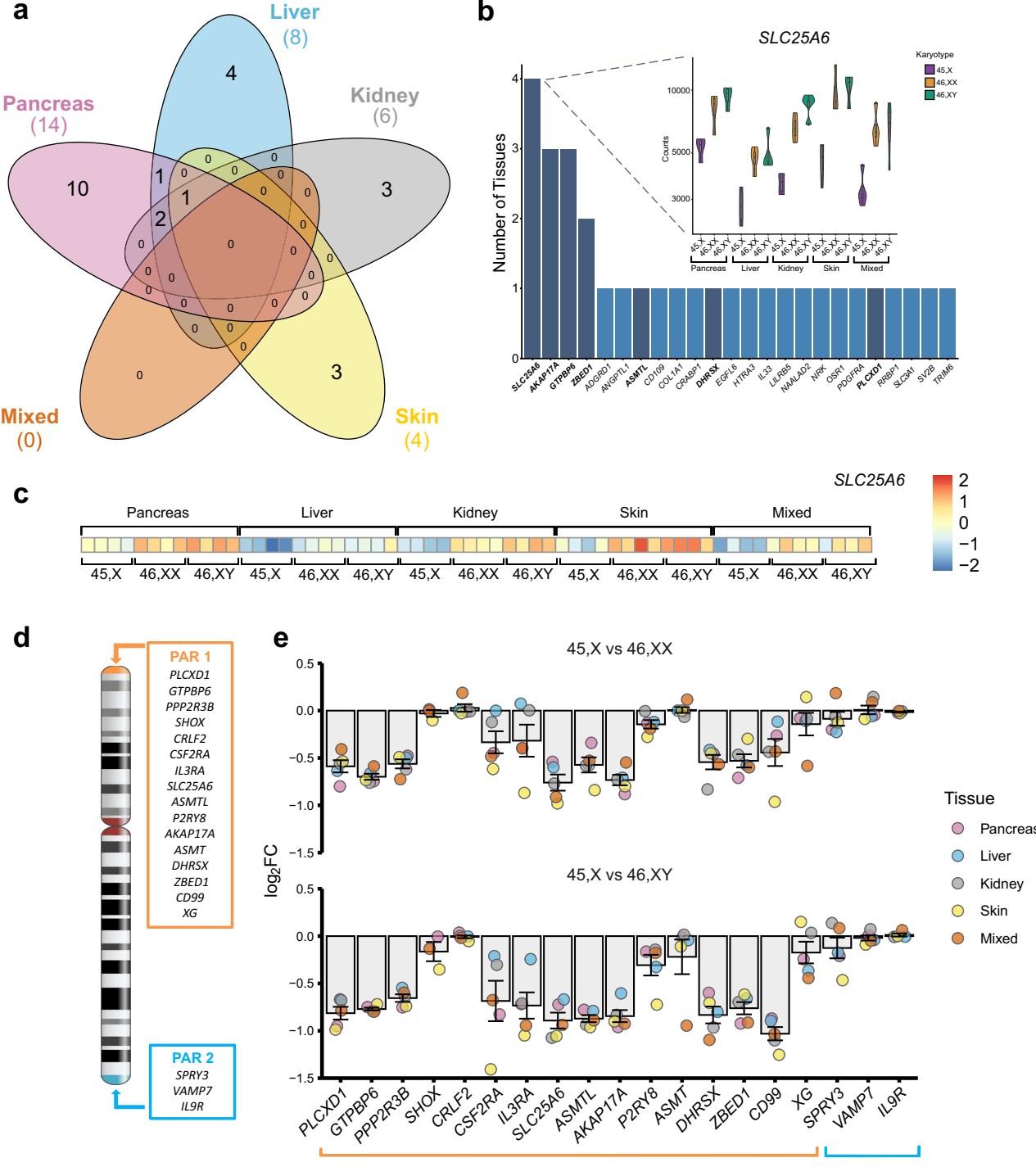

**Fig. 3 | Genes with lower expression in monosomy X when compared to 46,XX or 46,XY controls. a** Five-way Venn diagram showing overlap of genes with lower expression in monosomy X when compared to control karyotypes across all tissues. **b** Graph to show genes with lower expression in monosomy X across all tissues with inset of violin plot showing normalized count data of *SLC25A6* across all tissues (from panel **a**) according to karyotype (*n* = 4 in each group). Pseudoautosomal region (PAR) genes are shown in bold and dark bars. **c** Heatmap of *SLC25A6* expression across all tissues used in study. **d** Schematic of the X chromosome and showing PAR1 and PAR2 genes. **e** Differential expression of genes located on the PAR regions for 45,X compared to 46,XX or 46,XY samples. Individual mean data points for each tissue group are shown, as indicated in the legend. The bars represent the mean of the different tissue groups with standard error of the mean shown. $\text{Log}_2$ fold change (FC) = −1.0 represents half the expression in 45,X samples (i.e., haploinsufficiency), whereas $\text{log}_2\text{FC}$ = 0 represents similar expression in 45,X samples and 46,XX and 46,XY controls (*n* = 4 for each karyotype in each tissue group).

enrichment pathway analysis (*n* = 74 genes), biological functions emerged related to dissecting aneurysm, aneurysm of the ascending thoracic aorta and connective tissue disorders (Fig. 5b). These functions resulted largely from lower expression of core connective tissue genes in monosomy X (i.e.,

*FBN1*, *COL5A1*, *COL3A1*, *COL1A1*, *MMP2*, *COL5A2*, *COL1A2*) (Fig. 5c). The mean differences in gene dosage were subtle, and not all values were significant when adjusted for multiple comparisons, but more marked differences were seen in the skin where collagen genes are innately expressed

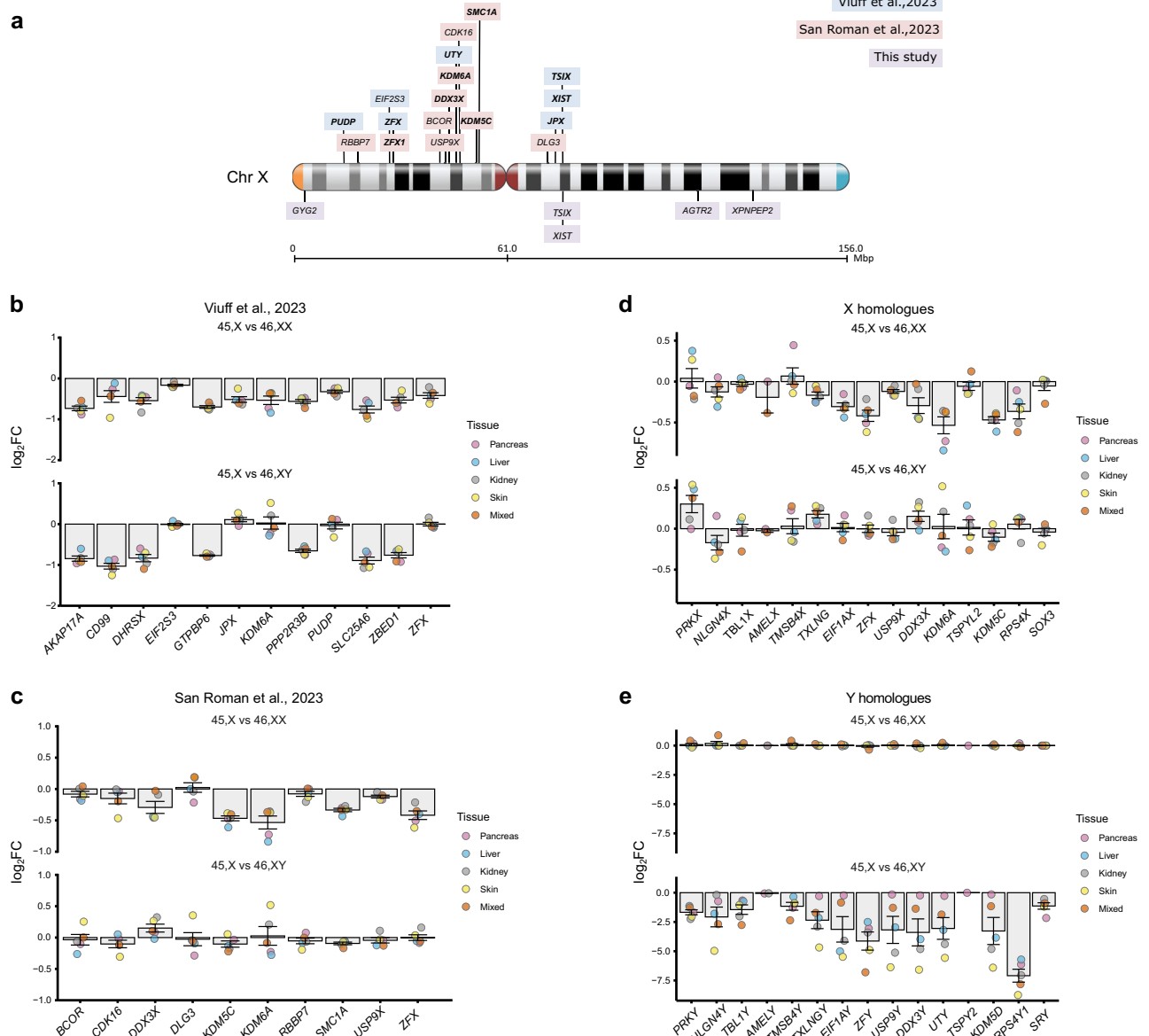

**Fig. 4 | Additional key genes proposed to have lower expression in monosomy X, including X-Y homolog pairs. a** Schematic showing the genomic location of key genes of interest on the X chromosome (see Fig. 4b, c). PAR genes are not shown. **b** Differential expression in our datasets of selected "core" X chromosome genes that have been linked to monosomy X, including those proposed to show X chromosome dosage (from Viuff et al. 2023). Individual mean data points for each tissue group are shown, as indicated in the legend. The bars represent the mean of these different tissue groups with standard error of the mean shown. Log$_2$ fold change (FC) = −1.0 represents half the expression in 45,X samples (i.e., haploinsufficiency), whereas log$_2$FC = 0 represents similar expression in 45,X samples and 46,XX and 46,XY controls (n = 4 for each karyotype in each tissue group). Note *XIST* and *TSIX* show strong differential expression in all datasets but are omitted from the graphic. **c** Differential expression in our datasets of selected X chromosome genes linked to "sex differences" (from San Roman et al.)[27]. Data are presented as described above. Note the different y-axis scale compared to Fig. 4b. **d** Differential expression in our datasets of key "ancestral X-Y homolog" gene pairs (from Godfrey et al. 2020)[37]. This panel shows the X chromosome gene of the pair, with the corresponding Y chromosome homolog gene below it (Fig. 4e). Data are presented as described above. **e** Differential expression in our datasets of key "ancestral X-Y homolog" gene pairs (from Godfrey et al. 2020)[37]. This panel shows the Y chromosome gene of the pair, with the corresponding X chromosome homolog gene above it (Fig. 4d).

(Fig. 5c). We hypothesize, therefore, that lower expression of these genes could confer a risk for the development and progression of aortic root dilatation and aortic dissection in women with TS, especially as loss of *FBN1* is found in Marfan Syndrome where thoracic aortic aneurysm can be a key feature[38].

**Genes with higher expression in monosomy X tissues**

Although lower expression of many X chromosome genes was expected in monosomy X tissues, especially in the PAR1 region, we also addressed whether any X chromosome or autosome genes show consistently *higher* expression in monosomy X tissues compared to matched controls.

In order to achieve this, a tissue specific analysis was first performed to identify genes that were higher in both 45,X *versus* 46,XX and 45,X *versus* 46,XY datasets (log$_2$FC > 0.5, p-adj ≤ 0.05; Supplementary Fig. 7). Next, the intersect of these genes across all five tissue groups was generated (Fig. 6a, b).

Remarkably, only one gene was consistently identified across all tissues (Fig. 6a). This gene, OVCH1 Antisense RNA 1 (*OVCH1-AS1*) had higher expression in all monosomy X tissues, including the mixed group, when

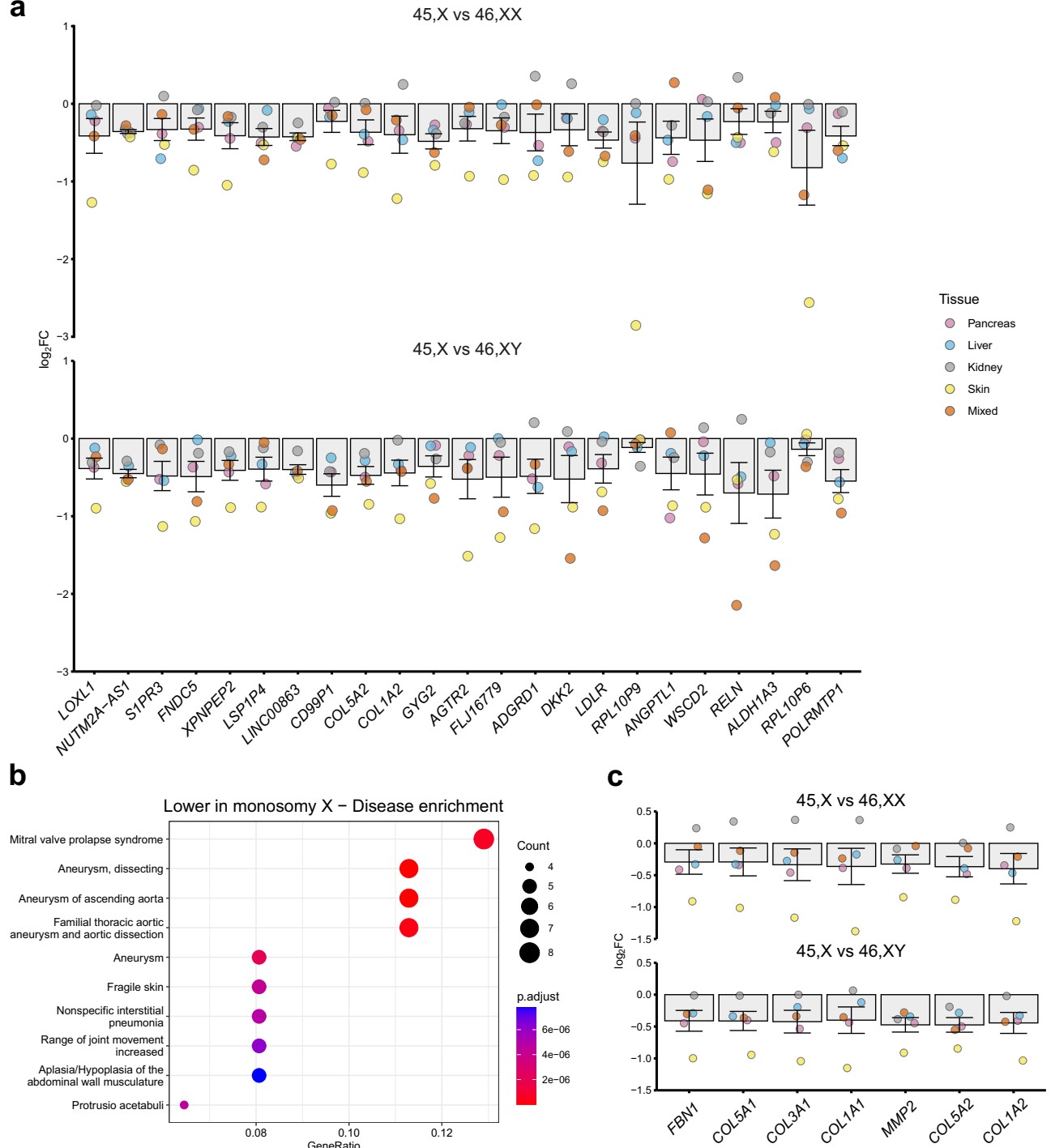

**Fig. 5 | Additional genes that have lower expression in monosomy X samples and related pathways. a** Differential expression data of other non-PAR X chromosome and autosome genes that are lower in 45,X compared to 46,XX or 46,XY samples. Individual mean data points for each tissue group are shown, as indicated in the legend. The bars represent the mean of these different tissue groups with standard error of the mean shown. Log2 fold change (FC) = −1.0 represents half the expression in 45,X samples (i.e., haploinsufficiency), whereas log2FC = 0 represents

similar expression in 45,X samples and 46,XX and 46,XY controls ($n = 4$ for each karyotype in each tissue group). Data are ordered according to the mean of the 45,X *versus* 46,XX and 45,X *versus* 46,XY datasets. **b** Dot plot showing disease enrichment analysis for genes lower in 45,X compared to mean controls ($n = 74$), with log2FC cut-off below −0.35. **c** Differential expression in our dataset of the key genes contributing to the thoracic aortic aneurysm disease enrichment pathway. Data are presented as described above. See Fig. 5a for legend.

compared to 46,XX and 46,XY controls (Fig. 6b−d; see also Fig. 2, Supplementary Fig. 7, and Supplementary Data 35 and 36). These findings were confirmed using qRT-PCR (Fig. 6e, f), and in replication datasets of bulk RNA-Seq from placental tissue and muscle (Fig. 6g, h).

*OVCH1-AS1* is a long non-coding RNA (lncRNA) gene located on chromosome 12 (UCSC, chromosome12:29,389,642-29,487,473, human GRCh38/hg38) (Fig. 7 a, b). *OVCH1-AS1* has been identified as a potential antisense transcript of the protein coding gene, ovochymase 1 (*OVCH1*),

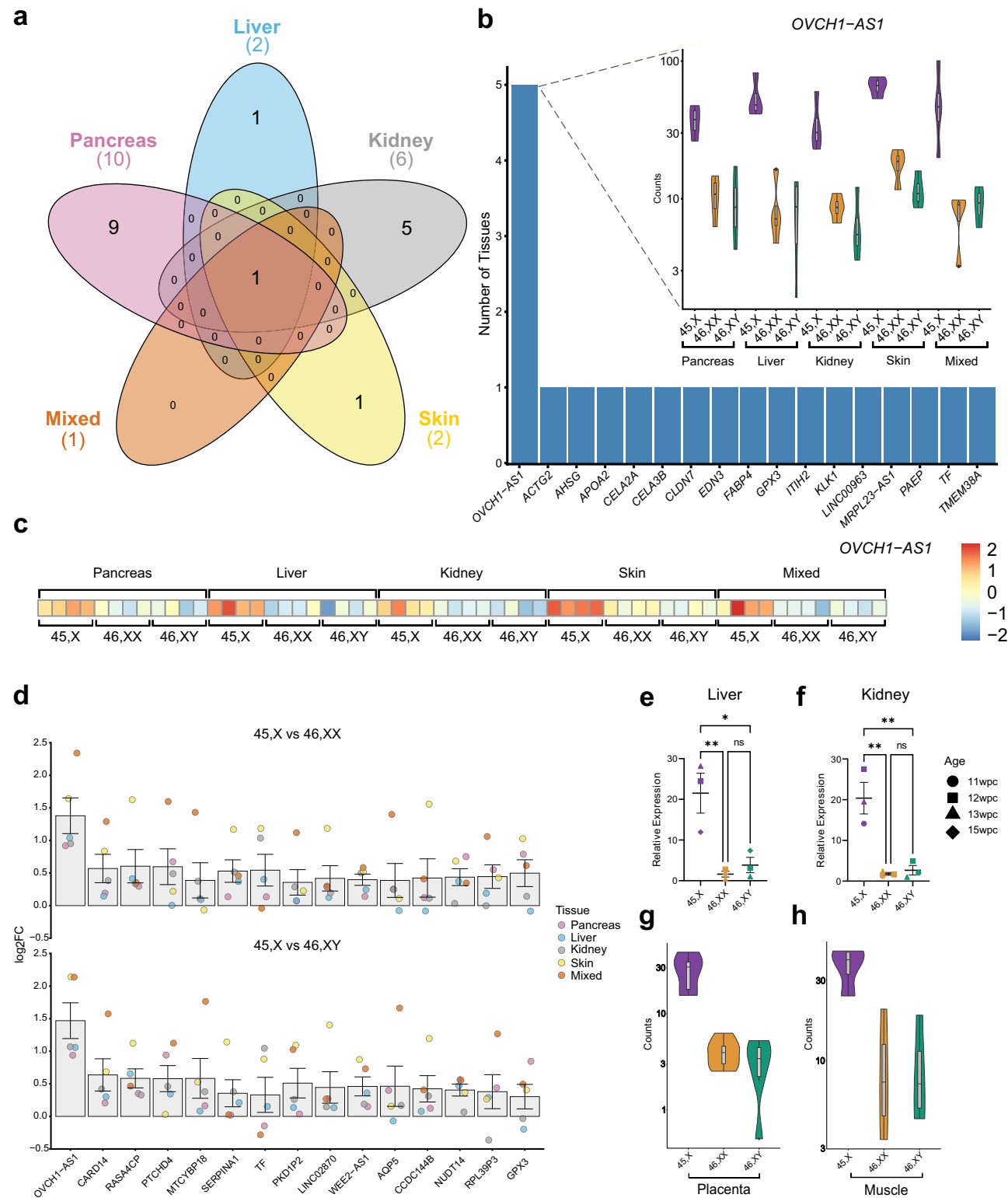

and maps to a locus containing *FAR2* (forward strand) and *TMTC1* and *ERGIC2* (reverse strand) (Fig. 7a).

Relatively little is known about *OVCH1-AS1*, but new studies are emerging[39]. In the GTEx database of adult tissue gene expression, *OVCH1-AS1* generally has low level expression in adult tissues, as is often the case with lncRNAs (Supplementary Fig. 8). Conservancy analysis in vertebrates using Multiz alignments (UCSC genome browser) showed that *OVCH1-AS1* is not highly conserved beyond higher primates; for example, it is not

identified in mouse (Fig. 7b). Although a potential open reading frame exists between nucleotides 231-825 (203 codons) using the Coding-Potential Assessment Tool (CPAT) (RNAcentral, transcript variant 1 (URS000075D789_9606; CPAT coding probability 0.824, cut-off of 0.364), most other algorithms suggest *OVCH1-AS1* (RefSeq ID: NR_073172) is a lncRNA rather than a protein coding gene (Supplementary Table 1).

As lncRNA and anti-sense RNA transcripts can potentially modulate gene transcription, we analyzed the relative expression of all chromosome

**Fig. 6 | Genes with higher expression in monosomy X when compared to 46,XX or 46,XY controls. a** Five-way Venn diagram of differentially expressed genes with higher expression in monosomy X tissues. **b** Graph to show genes with higher expression in 45,X with inset showing violin plot of normalized count data of *OVCH1-AS1* across all tissues according to karyotype (*n* = 4 in each group). **c** Heat map of *OVCH1-AS1* expression across all tissues used in the study. **d** Differential expression data of genes that are higher in 45,X compared to 46,XX or 46,XY samples. Individual data points for each tissue are shown, as indicated in the legend. The bar represents the mean of these data points with the standard error of the mean shown. Here, log$_2$ fold change (FC) = 1.0 represents twice the expression in 45,X tissues, whereas log$_2$FC = 0 represents similar expression in 46,XX and 46,XY control and 45,X samples. (*n* = 4 for each karyotype in each tissue group). **e** Quantitative

real-time PCR (polymerase chain reaction) showing relative expression of *OVCH1-AS1* in fetal liver (*n* = 4 samples in each group). Data are presented as scatter dot plots with each datapoint being the mean of triplicate experiments. The mean and standard error of the mean of these datapoints is also shown. **f** Quantitative real-time PCR showing relative expression of *OVCH1-AS1* in fetal kidney (*n* = 4 samples in each group). Data are presented as scatter dot plots with each datapoint being the mean of triplicate experiments. The mean and standard error of the mean of these datapoints is also shown. **g** Violin plots of *OVCH1-AS1* expression (normalized counts) in 45,X placenta compared to 46,XX and 46,XY controls (*n* = 6 in each group). **h** Violin plots of *OVCH1-AS1* expression (normalized counts) in 45,X muscle compared to 46,XX and 46,XY controls (*n* = 4 in each group). *, *p*-value < 0.05; **, *p*-value < 0.01; ns, not significant.

12 genes in our dataset (45,X *versus* 46,XX and 45,X *versus* 46,XY, independently). We did not identify consistently altered gene expression in this region (Fig. 7a–c). Next, more granular analysis of expression of genes in the *OVCH1-AS1* locus was undertaken (*OVCH1, ERCIG2, FAR2, TMTC1*) using bulk RNA-seq data for each tissue as well as qRT-PCR. No significant differences in the relative expression of these genes were observed between monosomy X samples and the 46,XX and 46,XY controls in these tissues studied (Fig. 7d, e, and Supplementary Fig. 9 a, b, and Supplementary Data 37). Thus, *OVCH1-AS1* is clearly a consistently upregulated autosomal long non-coding RNA associated with monosomy X in multiple tissues, but its potential biological role as a mediator of phenotypes in women with TS requires further investigation.

## Multi-tissue analysis
Given the relatively small sample size in each tissue studied (*n* = 4 for each karyotype), an additional "multi-tissue" analysis was undertaken that combined the pancreas, liver, kidney, skin and mixed groups (*n* = 20 for each karyotype). Despite having more power to detect small changes, a similar group of genes was identified that were consistently lower (*AKAP17A1, ASMTL, GTPBP6, PLCXD1, POLRMTP1, SLC25A6, ZBED1*) or higher (*OVCH1-AS1, PAEP*) in monosomy X tissues (log$_2$FC > 0.5, p-adj < 0.05) (Supplementary Figs. 4f, 7f; Supplementary Data 38-41). By correlating the log$_2$FC for 45,X *versus* 46,XX with the same gene in 45,X *versus* 46,XY datasets, the distribution and magnitude of differentially expressed genes could be clearly seen (Fig. 8; see Supplementary Figs. 10 and 11 for individual tissue correlation plots). As the multi-tissue analysis contains several different tissue samples from the same fetus (Supplementary Fig. 1a), further analysis was undertaken using a linear mixed effects model with individual as a random effect. No major differences were seen. Fewer significant results were found due to a reduction in false positives, as expected for an interaction test (Supplementary Data 42).

## Monosomy X and the developing placenta
Although monosomy X is the only chromosomal monosomy compatible with survival in humans, it has been estimated that most monosomy X pregnancies are lost[5–7]. Several hypotheses have suggested that this could be due to fetal anomalies (such as lymphedema or hydrops fetalis), defects in placental development and function, or mechanisms involving the maternal-fetal interface[5,40,41]. Detailed transcriptomic analysis of human monosomy X placenta is lacking, especially during the early stages of fetal development when loss of many monosomy X pregnancies occurs.

In order to investigate this further at a biologically relevant time point (late first trimester/early second trimester), samples were taken from monosomy X placentas (*n* = 6) between 11 and 15 wpc, together with matched 46,XX (*n* = 6) and 45,XY (*n* = 6) control placental tissues (Fig. 9a, Supplementary Fig. 12a).

As placental mosaicism for a fetal 46,XX cell line has been proposed as a mechanism that can "rescue" or modify placental disruption[41], we initially undertook SNP array analysis using DNA derived from two to four independent areas of each placenta in the 45,X group (*n* = 6, total *n* = 18 areas), as well as 46,XX and 46,XY controls. As expected, a low level 46,XX

component was identified in all samples (45,X; 46,XY), consistent with the presence of limited maternal decidual tissue (Supplementary Fig. 13). Strong enrichment of a 46,XX line was not seen in 45,X samples compared to 46,XY controls. These findings were validated by investigating the expression of *XIST* in bulk RNA-seq data from 45,X samples, where similar transcript levels to 46,XY controls were seen (Fig. 9b and Supplementary Fig. 12b). Taken together, these data suggest that placenta mosaicism for a 46,XX cell line is not common in 45,X placenta, at least at this stage of gestation.

We then analyzed bulk RNA-seq data to identify genes that were lower in 45,X placenta (*n* = 6) compared to controls, which could help to elucidate potential mechanisms of pregnancy loss associated with monosomy.

Global transcriptomic analysis using PCA showed that placental samples clustered together compared to control tissues as expected (Supplementary Fig. 14), and that within the placenta samples an influence of karyotype and age was seen (Fig. 9c). Volcano plots comparing 45,X with 46,XX and 46,XY placental samples again revealed the expected pattern of *XIST* and Y chromosome gene changes, with higher expression of *OVCH1-AS1* also seen in 45,X tissue (Fig. 9d, e, and Supplementary Data 43–46). To investigate placenta-specific genes that were consistently lower in 45,X placenta, a four-way analysis was undertaken using datasets derived from 46,XX or 46,XY *versus* 45,X placenta ("45,X lower" genes), as well as from 46,XX or 46,XY placenta *versus* 46,XX or 46,XY control tissues ("placenta-specific" genes) (Fig. 9f, Supplementary Data 47, 48). Using this approach, the PAR1 gene *CSF2RA* (UCSC, chrX:1,268,814-1,309,935 41,122, human GRCh38/hg38) was identified as the principal placenta-specific gene that is lower in monosomy X (Fig. 9f–i, and Supplementary Data 49). Lower expression of an autosomal gene *AADACL3* (UCSC, Chr1:12,716,110-12,728,760, human GRCh38/hg38) was also seen, although expression levels across samples were lower and more variable (Fig. 9f, h, i, and Supplementary Data 49).

*AADACL3* encodes arylacetamide deacetylase-like3, a potential membrane protein with hydrolase activity. This gene has low level expression in adult tissues and is only found in placenta, skin and breast (Human Protein Atlas Consensus RNA-seq) (Supplementary Fig. 15a). Consistent with this, analyses of available single cell RNA-sequencing (scRNA-seq) datasets[42–44] show low level expression of *AADACL3* in the extravillous trophoblast of early placenta, and in cytotrophoblasts, stromal cells, endothelial cells and decidual cells in late third trimester and pre-term tissue (Supplementary Fig. 16c and Supplementary Fig. 17b)[42–44]. The role of *AADAC3L* and its protein in the placenta is unknown.

In contrast, *CSF2RA* shows much stronger and more specific expression in placenta, as well as in adult haematopoietic/immune regulating tissues (bone marrow, lymph node, tonsil, spleen) (Human Protein Atlas Consensus RNA-seq) (Supplementary Fig. 15b). *CSF2RA* encodes the Colony Stimulating Factor 2 Receptor Subunit Alpha (also known as Granulocyte-Macrophage Colony-Stimulating Factor (GM-CSF) Receptor Subunit Alpha or CD116). This protein forms part of a heterodimeric cytokine receptor complex that mediates the effects of colony stimulating factor 2 (CSF2)[45]. CSF2 (GM-CSF) acts through this low affinity receptor via STAT5 to stimulate the proliferation, differentiation and functional activation of hematopoietic cells[45]. In available single cell RNA sequencing

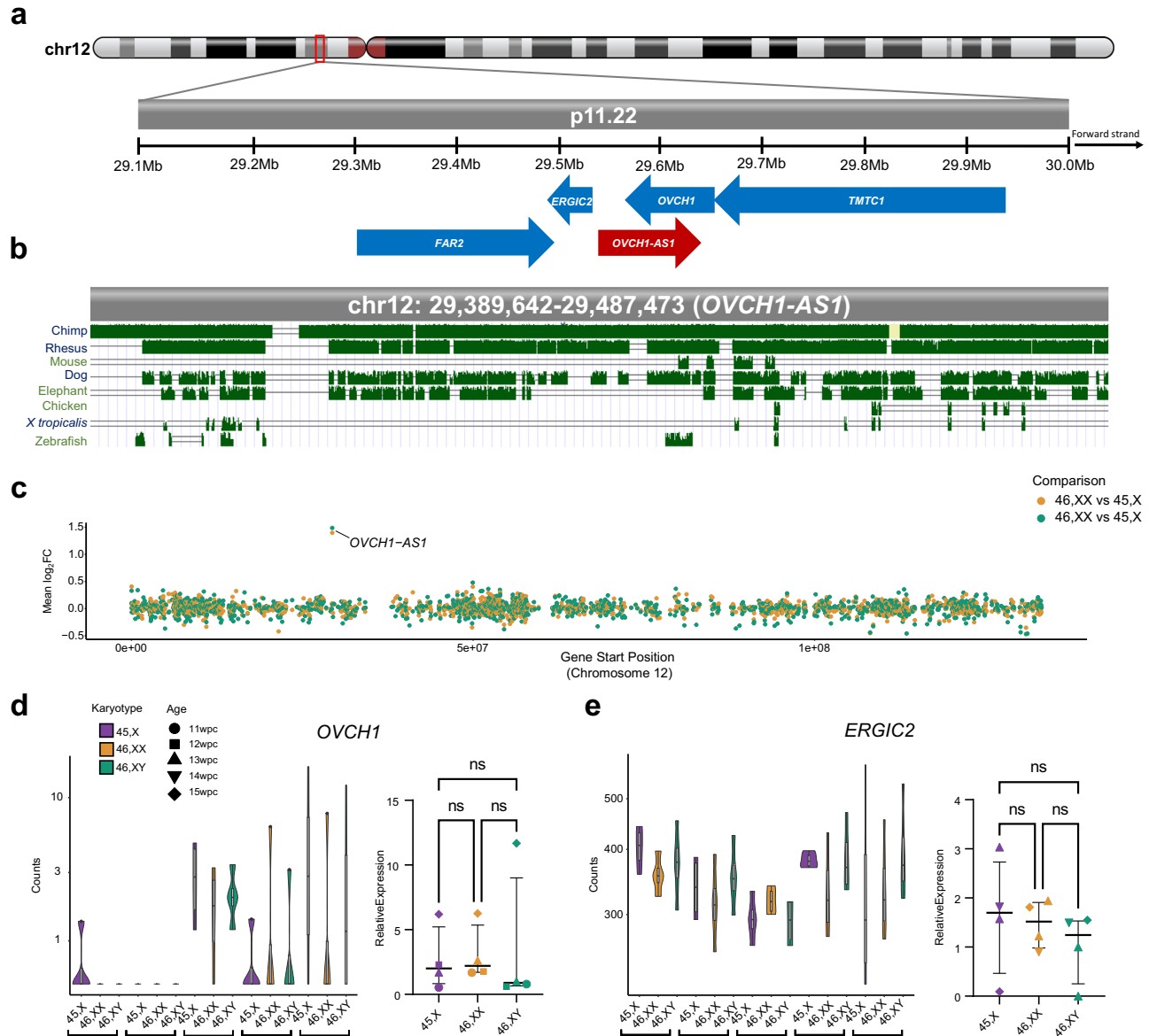

**Fig. 7 | OVCH1-AS1 locus and conservation. a** Overview of chromosome 12 (p.11.22) and genes located in the region of *OVCH1-AS1*. Arrows indicate direction of transcription. The red arrow indicates *OVCH1-AS1* (non-coding RNA gene) and blue arrows indicate protein coding genes. **b** Multiz alignments of species across the *OVCH1-AS1* locus. A single line indicates no transcript in the aligned species. Double lines show aligning species having one or more unalignable transcripts in the gap region. **c** Relative expression of all chromosome 12 genes in 45,X tissues compared to 46,XX (orange) and 46,XY (green). The higher expression of *OVCH1-AS1* in monosomy X tissue is indicated as log2 fold change (FC) in relation to control

tissues. **d** Violin plot of *OVCH1* in the different tissues (normalized counts from bulk RNA-sequencing) (left panel) (*n* = 4 samples in each group). Quantitative real-time PCR (polymerase chain reaction) showing relative expression of *OVCH1* in fetal kidney (right panel) (*n* = 4 samples each group). Data are presented as scatter dot plots with mean and standard error of the mean also shown. **e** Violin plot of *ERGIC2* in different tissues (bulk RNA-seq) (left panel) (*n* = 4 samples in each group). Quantitative real-time PCR showing relative expression of *ERGIC2* in fetal pancreas (right panel) (*n* = 4 samples each group). ns, not significant.

(scRNA-seq) data from first trimester placenta, *CSF2RA* expression was observed in placental villous cytotrophoblast, extravillous trophoblast and syncytiotrophoblast cells, and well as in decidual macrophages (Fig. 10a and Supplementary Fig. 16)[42]. More granular analysis of term placenta by scRNA-seq shows high expression of *CSF2RA* in many hematopoietic cell lineages (macrophages, NK-cells, activated and resting T-cells), as well as in non-proliferative interstitial cytotrophoblasts (Supplementary Fig. 17)[43].

To obtain more direct evidence for the role of *CSF2RA* in the monosomy X placenta, we undertook immunohistochemistry (IHC) for CSF2RA in 45,X placenta compared to 46,XX and 46,XY control samples. Between 11wpc and 15wpc the monosomy X placenta shows increasing

non-specific inflammatory changes as well as irregular villus tufting (Fig. 10b). CSF2RA localized to the syncytio- and cyto-trophoblastic layers in all samples as expected, but in the 45,X placenta at 14wpc showed less defined surface staining (Fig. 10c). Furthermore, pathway enrichment analysis of genes (*n* = 266; Supplementary Data 46) that are lower in monosomy X placenta compared to control (46,XY) identified processes linked to leukocyte activation and adhesion, immune response, and cytokine signaling (Fig. 10d).

Taken together, these data provide some of the first direct evidence that *CSF2RA*/CSF2RA is altered in monosomy X placenta, and these findings suggest that dysregulation of immune and inflammatory processes may

**Fig. 8 | Correlation plot of gene expression in 45,X tissues compared to 46,XX (y-axis) and 46,XY (x-axis) showing global patterns of differential changes.** Each data point represents a specific gene in each of five different tissue groups (pancreas, liver, kidney, skin, mixed group). Genes with higher expression in monosomy X are shown in the upper, right quadrant (e.g., *OVCH1-AS1*). Genes with lower expression in monosomy X are shown in the lower, left quadrant. Key clusters of Y chromosome genes and X-inactivation genes (e.g. *XIST*) are also shown.

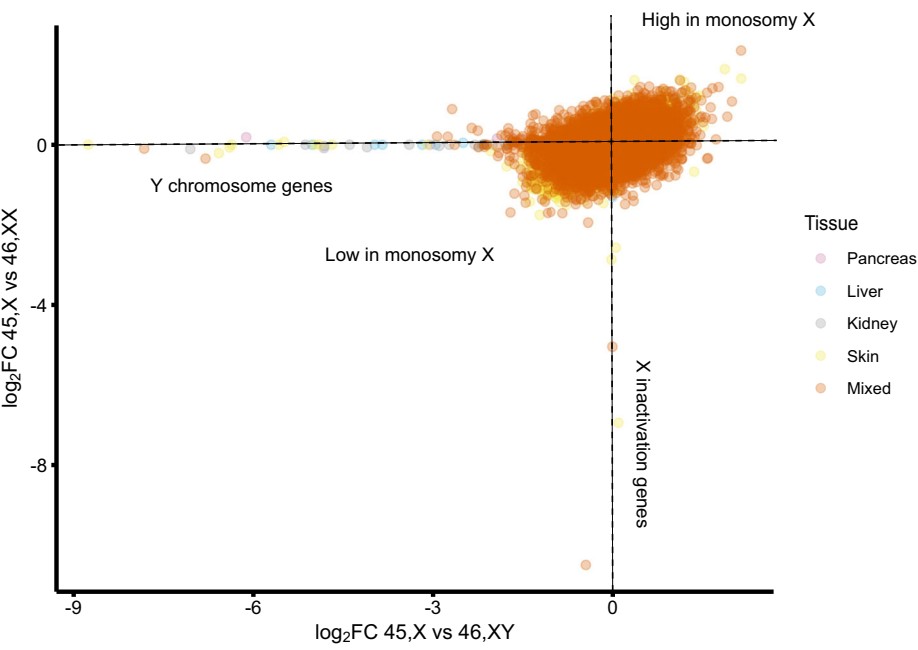

contribute to the placental dysfunction associated with loss of an X chromosome.

## Discussion

Monosomy X is the most common chromosomal aneuploidy in humans and can be associated with a broad range of clinical features in Turner syndrome. Some of these features have a developmental origin. The exact mechanisms that give rise to phenotypes in women with TS are still poorly understood, as are the mechanisms responsible for the high incidence of pregnancy loss. We therefore undertook a detailed transcriptomic analysis of monosomy X tissues during early human development, to better understand the molecular basis of Turner syndrome, with the ultimate aim of developing personalized approaches to long-term management.

We first assessed genes with lower expression in monosomy X, to be consistent with the long-standing hypothesis that haploinsufficiency of genes in the X chromosome may drive phenotypic features. As expected, many of these genes were pseudoautosomal genes (e.g., *SLC25A6*, *AKAP17A*, *GTPBP6*, *ZBED1*, *ASTML*, *DHRSX*, *PLCXD1*) expressed from the PAR1 region. Not all PAR genes showed lower expression, mostly due to lower background transcript levels or greater variability in expression between different tissues (e.g., *SHOX*). A lack of SHOX differential expression has been found previously[46]. Notably, we also found PAR2 genes had no differences in expression between monosomy X and control karyotypes. This finding has also been shown in other recent studies[20,21,32], suggesting that the dynamics of X chromosome PAR2 regulation may be different to PAR1.

Several other studies have looked for key X chromosome genes that have lower expression in adult Turner tissues[20,21,26,29,32,47,48]. These studies have mainly analyzed blood leukocytes and more recently muscle and fat, or derived cell lines[20,32–34]. In addition to several PAR1 genes, the study by Viuff et al.[20] found a subset of key genes that were differentially expressed by X chromosome dosage (45,X *versus* 46,XX and 47,XXY *versus* 46,XY), including genes that escape X inactivation (e.g., *PUDP*, *ZFX*, *KDM6A*, *JPX*, *TSIX* and *XIST*). These genes were also generally found to be differentially expressed in our dataset. Another recent study by San Roman et al.[27] proposed a core group of X chromosome genes that are linked to sex differences. Several of these genes were found to be lower in our datasets, such as *DDX3X*, *KDM5C*, *KDM6A*, *SMC1A* and *ZFX*. By analysing ancestral X-Y

gene pairs[37], we showed consistently lower expression in monosomy X for *EIF1AX/EIF1AY*, *ZFX/ZFY*, *DDX3X/DDX3Y*, *KDM6A/UTY*, *KDM5C/KDM5D* and *RSP4X/RSP4Y1*. Although these genes can influence transcription and translation, the role of histone demethylases (e.g., *KDM5C/KDM5D*, *KDM6A/UTY*) may be particularly important. For example, pathogenic variants in *KDM6A* cause Kabuki syndrome 2 (KABUK 2, X-linked) (OMIM 300867), which has a high prevalence of TS-associated features such as horseshoe kidney, short stature and hearing loss[49]. Here, we report, for the first time, core groups of PAR and XCI genes, and ancestral X-Y gene pairs that show lower expression in key biologically-relevant fetal monosomy X tissues such as pancreas, kidney, liver and skin, rather than the typical approach of studying adult samples, usually blood leukocytes.

Extending our analysis to other genes with lower expression (log₂FC < -0.35) in monosomy X identified several autosomal genes which may be clinically relevant. For example, lower expression of the low-density lipoprotein receptor gene (*LDLR*), which regulates cholesterol homeostasis, could contribute to the risk of dyslipidemia in Turner syndrome[50]. Previously, increased *LDLR* expression has been reported in amniotic fluid RNA in a small number of monosomy X fetuses[46]. Furthermore, pathway enrichment analysis identified biological processes related to ascending/thoracic aortic aneurysm and connective tissue disease[38,51]. Aortic root dilatation and thoracic aortic aneurysm (TAA) is rare in the population but can be a significant risk in women with TS[17,52]. Aortic root dilatation is now regularly monitored in adulthood[3,16,18,51] and TAA has the highest odds ratio for mortality in the adult TS group (standardized mortality ratio, =23; 95% CI 13.8–37.8)[3,53]. The main genes identified in this pathway were collagen (*COLA*) genes and fibrillin 1 (*FBN1*). Defects in *FBN1* cause Marfan syndrome, which is also associated with TAA[38]. This finding raises the possibility of a pathogenic link between the TS and Marfan syndrome. Further studies with larger sample numbers, and looking more specifically at developing aortic tissue, are needed.

We also hypothesize that this risk of TAA could be compounded by lower expression of *AGTR2* (angiotensin II receptor type 2, AT2R) in monosomy X. *AGTR2* typically antagonizes the effects of the angiotensin II type 1 receptor (AT1R). Lower *AGTR2* could lead to higher blood pressure and vascular proinflammatory effects[54]. Several studies have shown higher plasma renin activity in women with TS[55,56]. In addition, monosomy X augments the severity of angiotensin II-induced aortopathies in mice[57].

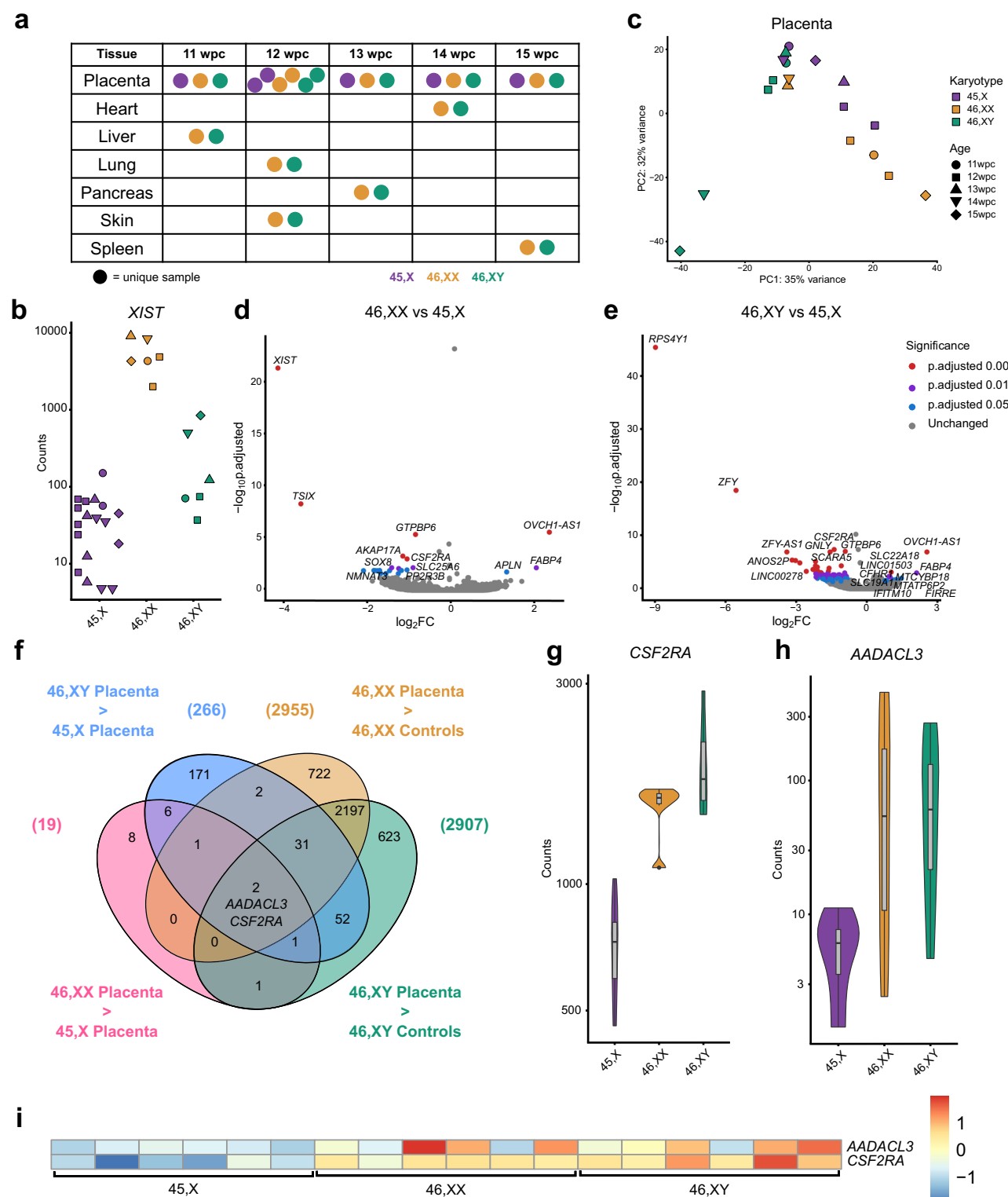

Notably, AT1R blockade has been shown in recent systematic reviews to reduce the risk of thoracic aortic aneurysm in Marfan syndrome[58]. Further studies are needed to assess whether the interplay of these factors could lead to increased TAA risk in women with TS and be an avenue for new therapeutic approaches.

Although most of the effects of TS might be expected to be due to a reduction of gene dosage, one gene (*OVCH1-AS1*) was consistently higher in all datasets studied. *OVCH1-AS1* encodes a long non-coding RNA that is

expressed at relatively low transcript level, which is typical for lncRNAs, but with at least two-fold increase in expression in all 45,X tissues assessed. The potential role of *OVCH1-AS1* is not known. Recent studies have suggested different expression levels of leukocyte *OVCH1-AS1* in relation to frailty with age in humans[39], as well as higher OVCH1-AS1 expression in biopsies from Crohn's disease[59]. Furthermore, several studies have identified *OVCH1-AS1* as a higher-expressed gene in leukocytes, fat, and muscle from adult women with TS [20,21,26,29,47]. This finding was rarely highlighted in detail,

**Fig. 9 | Differential expression of genes in the 45,X placenta compared to 46,XX or 46,XY controls. a** Overview of placental age stages used for the study. Karyotypes are indicated by the key. wpc, weeks post conception. **b** Expression of the X inactivation regulator *XIST*, bulk RNA-sequencing (RNA-seq) counts in the 45,X placenta group ($n = 18$) compared to 46,XX placenta ($n = 6$) and 46,XY placenta ($n = 6$). **c** Principal component analysis (PCA) of 45,X, 46,XX and 46,XY placental samples used in the study. PC, principal component. **d** Volcano plot showing differential expression of genes between the 45,X and 46,XX matched control samples ($n = 4$ in each group). The top ten most differentially expressed genes in each dataset are labeled, based on adjusted p-value (p-adj) and where $\log_2$ fold change (FC) is greater than $+/-0.5$. Genes with higher expression in 45,X samples have a positive $\log_2$FC

and those with higher expression in control tissues have a negative $\log_2$FC. The significance level of highlighted points is shown in the key. **e** Volcano plot showing differential expression of genes between the 45,X and 46,XY matched control samples. **f** Venn diagram with the common intersection showing placenta-specific genes that are consistently lower in 45,X. Data were generated for each group ($n = 6$ *versus* $n = 6$) with a differential expression cut-off of $\log_2$FC > 0.5 and p-adj < 0.05. **g** Violin plot of *CSF2RA* expression (normalized counts) in the placenta (bulk RNA-seq) ($n = 6$ each group). **h** Violin plot of *AADACL3* expression (normalized counts) in the placenta (bulk RNA-seq) ($n = 6$ each group). **i** Heat map of *AADACL3* and *CSF2RA* expression across all placental samples ($n = 6$ samples in each group).

but here we have conclusively demonstrated that *OVCH1-AS1* is higher in monosomy X in 14 different sub-studies from multiple different tissues.

As antisense lncRNAs sometimes modulate sense strand RNA, or genes in the region, we undertook more detailed analysis of the *OVCH1-AS1* locus of chromosome 12 to investigate potential *cis* effects, but we did not see an obvious effect on the expression of *OVCH1* nor of other local genes. Of note, like many lncRNAs, this gene is present in humans but not in the mouse. We hypothesize that, if *OVCH1-AS1* is biologically relevant, this feature could contribute to marked differences in phenotype between monosomy X in humans and the murine model of a single X chromosome[60].

Some of the biggest questions to address are how monosomy X is such a common cause of pregnancy loss and also whether there are any specific features in those pregnancies that survive to term? It has been estimated that many monosomy pregnancies are lost mostly towards the end of the first trimester and that this contributes to a substantial proportion of spontaneous miscarriage in the general population[5,41,61].

One long-standing hypothesis is that term monosomy X pregnancies have placental mosaicism for a fetally-derived 46,XX line, but direct data are limited[41,62]. Here, we did not identify significant 46,XX mosaicism using both RNA (*XIST* counts) and DNA (SNP arrays) technologies in the six 45,X placental samples studied, suggesting that placenta mosaicism is not likely to be a common cause of fetal rescue in TS, at least at this gestational age, although placental sampling was limited to between two to four distinct regions of each 45,X placenta. Furthermore, because the placental material was obtained following termination in the early second trimester, we do not know whether spontaneous loss might still have occurred. Using a trophoblast-like model derived from X-monosomic human induced pluripotent stem cells, Ahern et al. showed impaired secretion of placental growth factor and human chorionic gonadotropin[63]. We did not observe significantly lower expression of the genes encoding these growth factors in our datasets. Nevertheless, whether there could be a tropic growth advantage for any small mosaic population of 46,XX cells to expand with time throughout gestation remains to be seen.

Another question relates to the mechanistic basis of pregnancy loss. Several studies have proposed that PAR genes such as *CSF2RA* are implicated, but direct data are limited[2,7]. We therefore developed a systematic approach to identify placenta-specific genes with lower expression in monosomy X placenta compared to control placenta. We found two genes of interest: *AADACL3* and *CSF2RA*.

*AADACL3* encodes arylacetamide deacetylase-like 3, a putative membrane hydrolase expressed at low level, primarily in the extravillous trophoblast. *AADACL3* expression was more variable between monosomy X samples and controls. The functional role of this gene in the placenta is unclear, although higher *AADACL3* expression has been reported in 46,XY preterm and term placental tissue compared to 46,XX[64].

More importantly, the PAR1 gene *CSF2RA* was observed in our dataset as having consistently lower expression in monosomy X compared to both 46,XX and 46,XY control placenta samples. In scRNA-seq analysis, *CSF2RA* is clearly expressed in early trimester and term placenta, as well as in macrophages, NK-cells and activated and resting T-cells, consistent with its expression in placenta and hematopoietic system-derived tissues in adult

panels[42,43,65]. CSF2RA encodes the alpha subunit of the CSF receptor, which mediates the effects of CSF2 (GM-CSF) on hematopoietic differentiation, immunity, and inflammation. Immune regulation within the fetal-placental interface is emerging as a key mechanism in pregnancy maintenance, and *CSF2RA* has been proposed as a candidate gene for early lethality in 45,X embryos[2,7]. *CSF2RA* has been shown to have 10-fold lower expression in 45,X-derived human embryonic stem cells compared to 46,XX-derived controls, and reduced expression in BDCA4+ dendritic cells from women with TS[7,66]. *CSF2RA* is implicated in placental development[67], and studies in bovine cultured embryos suggest CSF2 promotes embryo survival through antiapoptotic effects and by signaling of developmental pathways[68]. However, direct analysis of monosomy X placenta has not been reported. Of note, *Csf2ra* is autosomal in the mouse (chromosome 19).

Here, we show morphological changes in the monosomy X placenta during development. Using IHC, we have shown that CSF2RA localizes to syncytial trophoblast and basal trophoblast cells, with more diffuse patterning in monosomy X placenta at 14wpc at a time of degeneration of the brush border. Coupled with this, pathway analysis of lower-expressed genes in the monosomy X placenta showed enrichment of key biological processes, such as leukocyte activation, immune responses, and inflammation, involving key genes such as XCR1, interleukin 1 and interleukin family receptors. Taken together, these findings suggest that CSF2RA is likely to be a key modulator of fetal-placental dysfunction in monosomy X pregnancies and might have implications for our understanding of pregnancy loss in general. Further work to elucidate the potential mechanistic role of *CSF2RA* is needed, such as in model systems of monosomy X placental development using organoids, and in mediating immune defence mechanisms. Furthermore, as we studied placental material at 11-15wpc, we do not know if the samples we analysed harbored any specific features that protected against pregnancy loss at an earlier stage (first trimester), or whether the histological and transcriptomic changes seen could still have been associated with pregnancy loss at a later gestational age.

This study has several limitations. Firstly, access to tissues of interest was limited so only a relatively small number of samples were available and within a relatively narrow developmental time window, and one 45,X pancreas sample (13 wpc) was bisected in order to have four samples in this group. However, having samples within a 4-week period and a carefully balanced study design reduced marked variations due to organ development, increasing the power to detect more subtle sex chromosome-related effects. Furthermore, a combined "multi-tissue" analysis provided additional data. Second, one fetus had low level mosaicism for a Y chromosome line. The overall effects of this variability on our findings were likely to be minimal. Third, alternative approaches such as scRNA-seq might have provided more granular detail of single cell transcriptomics, but risked sampling bias and limited detection of low-level RNA transcripts. Thus, a bulk RNA-seq strategy was used with robust experimental design, replicating data with both 46,XX and 46,XY control groups, and validating findings in multiple different tissue groups. This is important as gene dosage changes linked to loss of a single X chromosome are generally subtle and of a lower magnitude than in most studies of developmental differential gene expression. Finally, sampling a greater number of regions of the placenta would have provided more data to exclude significant 46,XX confined

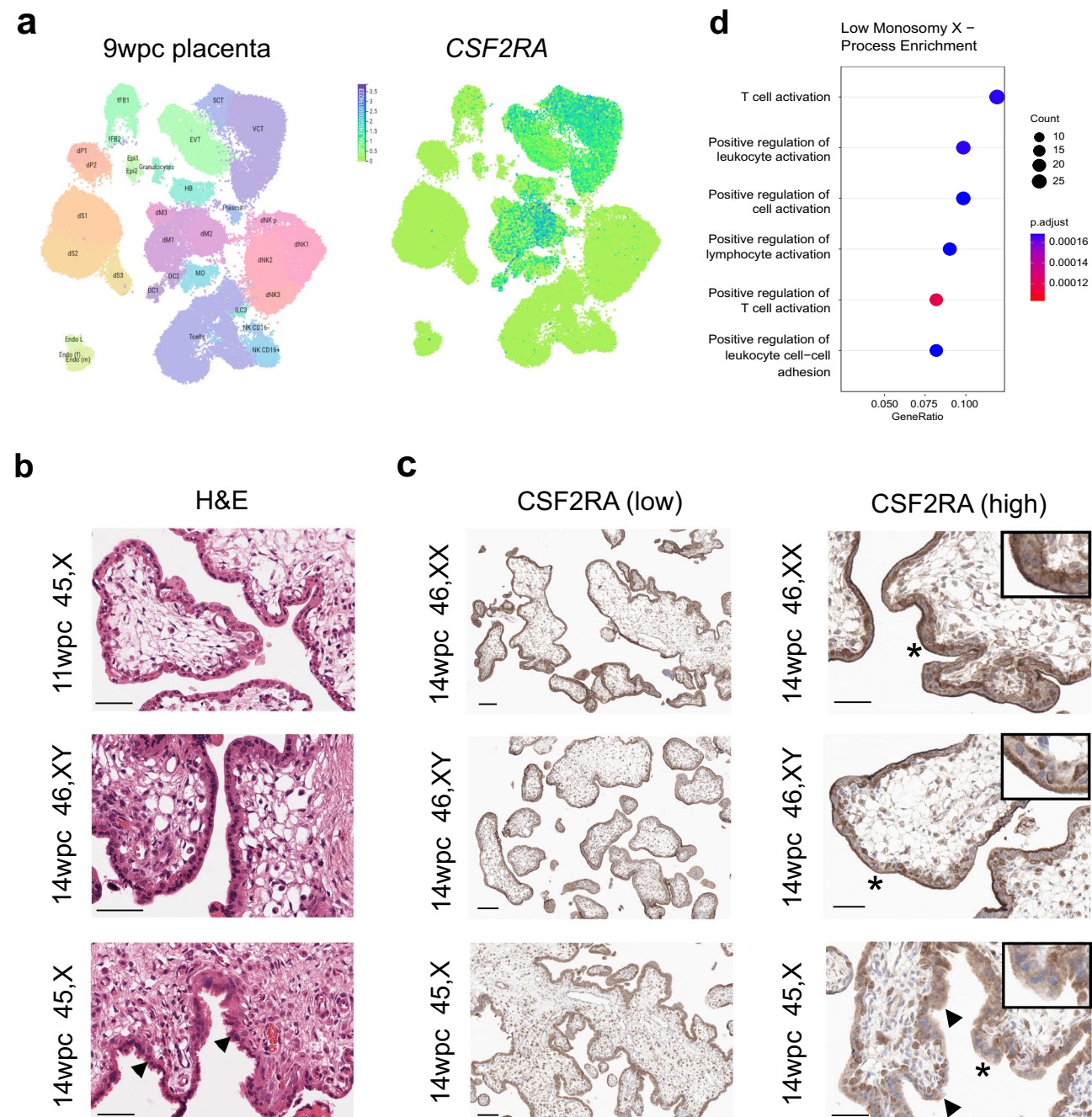

**Fig. 10 | Expression of *CSF2RA*/CSF2RA in the placenta and related pathways.**
**a** Consensus first trimester single cell data Uniform Manifold Approximation and Projection (UMAP) for clusters based on cell type (*left panel*) and feature plot for *CSF2RA* (*right panel*). These data were generated by the Vento-Tormo/Teichmann groups at the Wellcome Sanger Institute, Hinxton, UK and can be accessed using CZ CELLxGENE from https://maternal-fetal-interface.cellgeni.sanger.ac.uk/ (Vento-Tormo, R., Efremova, M., Botting, R.A. et al. Single-cell reconstruction of the early maternal–fetal interface in humans. Nature 563, 347–353 (2018); https://doi.org/10.1038/s41586-018-0698-6)[42]. This graphic is generated under a Creative Commons Attribution-BY 4.0 International License (https://creativecommons.org/licenses/by/4.0/). DC dendritic cells, dM decidual macrophages, dS decidual stromal cells, Endo endothelial cells, Epi epithelial glandular cells, EVT extravillous trophoblast, f fetal, F fibroblasts, HB Hofbauer cells, ILC innate lymphocyte cells, l lymphatic, m maternal, M3 maternal macrophages, PV perivascular cells, p proliferative, SCT

syncytiotrophoblast, VCT villous cytotrophoblast. **b** H&E staining of 45,X placenta at 11 weeks post conception (wpc) (*upper panel*), 46,XY placenta at 14 wpc (*center panel*) and 45,X placenta at 14 wpc) (*lower panel*) (scale bars 50 μm). Arrowheads show regions of irregular villus border in the 14wpc 45,X sample.
**c** Immunohistochemistry of CSF2RA (Colony Stimulating Factor 2 Receptor Sub-unit Alpha) in 46,XX, 46,XY and 45,X placenta at 14wpc at lower magnification (*left panels*) (scale bars 100 μm) and at higher magnification (*right panels*) (scale bars 50 μm). Arrowheads show regions of irregular border and diffuse CSF2RA staining in the 14wpc 45,X villus. Inset images show higher magnification of regions indicated by an asterix. **d** Biological process enrichment analysis for genes lower in 45,X placenta compared to 46,XY controls (bulk RNA-sequencing $n = 6$ each group; 266 genes identified) with a differential expression cut-off of $\log_2FC > 0.5$ and adjusted $p$-value < 0.05. The number of differentially expressed genes in 45,X placenta compared to 46,XX controls was insufficient for pathway analysis.

placenta mosaicism, although none was seen in the 18 samples analysed from 6 placentae in the study.

To the best of our knowledge, this is the first detailed transcriptomic study of human monosomy X in key clinically relevant tissues (e.g., pancreas, kidney, liver, skin, and placenta). This work has highlighted several important findings with implications for sex chromosome biology, the underlying mechanisms that could contribute to TS and potential targeted therapeutics. These findings will be important for facilitating research in the future.

## Materials And Methods

### Tissue samples
Human fetal tissue samples included in this study were obtained with informed consent and with ethical approval (REC references: London-Fulham Research Ethics Committee, 18/LO/0822; North East – Newcastle and North Tyneside 1 Research Ethics Committee, 18/NE/0290) from the Medical Research Council (MRC)/Wellcome Trust-funded Human Developmental Biology Resource (HDBR) (http://www.hdbr.org). The HDBR is a biobank regulated by the U.K. Human Tissue Authority (HTA) (www.hta.gov.uk). All ethical regulations relevant to human research participants were followed. Samples were obtained following elective termination, and initially stored in Leibovitz's L15 media (Gibco/Thermo Fisher Scientific, Waltham, MA, USA) on ice (mean <2 h) before being frozen (-70 °C) or transferred to formalin. Fetal age was estimated by measuring foot-length and knee-heel length linked to standard growth data. Fetal karyotype assessment was undertaken using quantitative PCR (chromosomes 13, 15, 16, 18, 21, 22 and X and Y) and confirmed with array analysis when aneuploidy was suspected. Tissues with a monosomy X (45,X) karyotype were stored at -70 °C. Control samples (46,XX, 46,XY), matched for tissue and age, were also obtained from the HDBR. Those with any other chromosomal anomaly or developmental features were excluded. An overview of samples is shown in Supplementary Data, SD1.

### Study design
A range of tissues, that are potentially relevant to the clinical phenotypes in Turner syndrome, were used for transcriptomic analysis. These included pancreas (diabetes mellitus), liver (liver enzyme elevation), kidney (developmental anomalies, renal dysfunction) and skin (naevi), as well as a "mixed group" that was comprised of brain, spleen, lung, and heart (Fig. 1a). Each sequenced group consisted of four monosomy 45,X samples, with four age and tissue matched 46,XX control samples and four age and tissue matched 46,XY control samples, between 11 and 15 wpc. Because of limited tissue availability for the 45,X pancreas group, a 13 wpc sample was bisected sagittally from the head of the pancreas to the tail, so that RNA could be independently extracted from two halves of the same biological sample. These two parts showed a degree of biological variability similar to other pancreas samples (Fig. 1c), suggesting that the shared origin of the samples was not a major driver of the transcriptomic profile, and that including both samples in the analysis to obtain n = 4 was justified. Each organ in the mixed tissue group (brain, spleen, lung, and heart) was collected at a different age stage within the timeframe (11 to 15 wpc), to give a further perspective on other key tissues (Fig. 1a). A larger "multi-tissue" group was generated (n = 20 each karyotype), by combining data from these samples (pancreas, kidney. liver, skin, mixed group). An addition group of muscle samples (n = 4 each karyotype) was included for further validation of *OVCH1-AS1* expression.

For the placental transcriptomic study, six monosomy 45,X placenta samples were obtained together with six age matched 46,XX control placentae and six age matched 46,XY control placentae, between 11 and 15wpc. Tissues were stored in −70 °C until processing. For all 45,X, 46,XX and 46,XY placentae, the entire placental tissue (approximately 1–2 cm diameter) was visualized on ice and a sample (20–30 mg) was taken for immediate RNA/DNA extraction. For the six 45,X placentae, rapid dissection of between one (n = 3) and three (n = 3) further samples was performed from other distinct and distanced regions of the same placenta, for the assessment of potential 46,XX confined placental mosaicism. Thus, a total of 18 distinct 45,X samples were processed from six different 45,X

placentae (n = 3, two samples; n = 3, four samples). For histology and IHC, independent placental tissues were stored in 10% formalin.

### RNA/DNA extraction
Samples were removed from −70 °C storage and dissected further, where necessary, using a standardized approach so that the required amount of tissue was available (up to 30 mg). RNA and DNA extractions were performed using an AllPrep DNA/RNA Mini Kit following the manufacturer's protocol (Qiagen, Hilden, Germany). Tissues were immediately cut into 1mm³ pieces on ice and transferred to lysis buffer supplied (buffer RLT). Samples were dissociated in this lysis buffer using a Kimble™ Kontes™ motorized pellet pestle (DWK Life sciences, Mainz Germany). To isolate RNA, the first flow through from the AllPrep column was digested using proteinase K, subsequent ethanol washes undertaken to allow binding of total RNA including miRNA to the RNeasy mini spin column, followed by Dnase I digestion and further washes to ensure elution of high-quality RNA. RNA quality was assessed using a Tapestation 4200 platform (Agilent Technologies, Santa Clara, CA, USA). RNA integrity numbers (RINs) are shown in Supplementary Fig. 1. DNA was also extracted from the same samples using the manufacturer's protocol. In brief, homogenized lysate was passed through an AllPrep DNA mini spin column to bind genomic DNA. Following proteinase K digestion under optimised buffer conditions, the column was washed, and DNA eluted and quantified using NanoDrop spectrophotometer (Thermo Fisher Scientific, Waltham, MA, USA).

### SNP array analysis
In order to determine sex chromosome complement and potential mosaicism in all samples, SNP array analysis was undertaken on extracted DNA samples, The Illumina Global Screening Array platform was used (v3.0) (Infinium HTS Assay Reference Guide (# 15045738 v04) (Illumina, Inc. San Diego, CA, USA). Output data were analyzed in Illumina Genome Studio version 2.0. X chromosome mosaicism was assessed using methods described by Conlin et al.,[69].

### Bulk RNA library preparation and sequencing
For bulk RNA-seq studies, RNA was used to generate cDNA libraries using a KAPA mRNA HyperPrep Kit (Roche, Basel, Switzerland) on a Hamilton Starlet robotic platform (Hamilton Company, Reno, NV, USA) and quality was analyzed using a Tapestation 4200 platform (Agilent Technologies, Santa Clara, CA, USA). Libraries were subsequently sequenced on a NovaSeq S2 Flowcell (paired end 2x56bp) (Illumina). All samples in this study were prepared and sequenced at the same time, to avoid potential batch effects. Fastq files were processed by FastQC and aligned to the human genome (Ensembl, GRCh 38.86) using STAR (v2.7.3a). The matrix containing uniquely mapped read counts was generated using featureCounts part of the R package Subread.

### Bulk RNA-seq data analysis and data representation
The following analysis was performed in R (version 4.2.2). Comparison of RNA-seq sample dimensionality was undertaken for all samples and for each individual tissue group using principal component analysis (PCA) and plots were generated using ggplot2[70,71]. Pairwise differential-expression analysis of tissues related to karyotype was performed using DESeq2[72], comparing either four 45,X samples with four 46,XX samples, or four 45,X samples with four 46,XY samples. A similar multi-tissue linear analysis was undertaken (n = 20 each karyotype) in DESeq2, as well as using a linear mixed effects model in limma-voom, with individual as a random effect. For the placental study, six samples were used in each group. Data were generated as log₂ fold change (FC) differences between groups > 0.5 and were considered statistically significant with an adjusted p-value < 0.05. Volcano plots were generated using ggplot2. Heatmaps for differentially expressed genes in 45,X tissue compared to control 46,XX and 46,XY controls were generated using the pheatmap library in R. Violin plots of normalized counts of tissues were plotted in ggplot2 in R. Venn diagram analyses between different subsets of expressed genes was undertaken using Venny

2.1 or InteractiVenn[73,74]. Pathway analysis and enrichment analysis was undertaken using clusterProfiler[75], Metascape[76] and gProfiler[77] version: e110_eg57_p18_4b54a898.

### OVCH1-AS1 gene locus and conservancy

The *OVCH1-AS1* (*OVCH1 Antisense RNA 1*) gene locus (including neighboring genes) was defined using the University of California, Santa Cruz (UCSC) Genome Browser (human GRCh38/hg38; Chromosome12:29,389,642-29,487,473) and the Ensembl browser 110 (Human GRCh38.p13; Chromosome 12: 29,387,326-29,489,451). Conservation of the *OVCH1-AS1* locus was evaluated in the UCSC genome browser (Multiz alignments of 100 vertebrates) using default species settings (Chimp, Rhesus monkey, Mouse, Dog, Elephant, Chicken, *Xenopus tropicalis* and Zebrafish).

### Quantitative real-time PCR (qRT-PCR)

cDNA was generated from pancreas, liver and kidney RNA using SuperScript III reverse transcriptase (Thermo Fisher Scientific Inc, MA, USA). Quantitative-real-time polymerase chain reaction (qRT-PCR) was performed using TaqMan Fast Advanced MasterMix (Applied Biosystems MA, USA) and Taqman gene expression assays for *OVCH1-AS1* (Hs04333030_m1) and genes in the surrounding locus (*OVCH1* (Hs07289759_m1), *ERGIC2* (Hs00275449_m1), *FAR2* (Hs00216461_m1), *TMTC1* (Hs00405786_m1)) (Applied Biosystems, MA, USA). Analysis was carried out using the comparative CT ($2^{-\Delta\Delta CT}$) method[78]. *ACTB* (Hs01060665_g1) or *GAPDH* (Hs02786624_g1) were used as a housekeeping genes. Data were normalized to 13wpc 46,XY samples for each organ. The gene *OVCH1-AS1* was assessed in triplicate on three independent occasions and mean $2^{-\Delta\Delta CT}$ was obtained for each study. Genes in the region of *OVCH1-AS1* (*OVCH1, ERGIC2, FAR2* and *TMTC1*) were assessed in triplicate on one occasion. Data were plotted using GraphPad Prism (version 9.5 for Windows, GraphPad Software, San Diego, CA, USA, www.graphpad.com). Differences between groups were analyzed using one-way ANOVA (Kruskal-Wallis) (GraphPad Prism).

### In silico assessment of *OVCH1-AS1*

A systematic review of publications citing *OVCH1-AS1* was undertaken in PubMed and GoogleScholar, using the gene name as a search term (October 2023). Although *OVCH1-AS1* is considered a long non-coding RNA (lncRNA) (GeneCards GC12P029389, Ensembl ENSG00000257599), the potential to generate a protein coding transcript was analyzed using the following algorithms: CPAT coding probability, PhyloCSF score, PRIDE reprocessing 2.0, Lee translation initiation sites, Bazzini small ORFs (accessed via RNAcentral and Lncipedia, October 2023). Gene expression of *OVCH1-AS1* and exon usage in adult tissues was obtained from GTEx (https://www.gtexportal.org/home/gene/OVCH1-AS1; ENSG00000257599.1; exon usage Source symbol;Acc:HGNC:44484).

### Placenta immunohistochemistry (IHC)

Placental samples for staining were fixed in 10% formalin before being processed, wax embedded and sectioned (3 μm). Hematoxylin and eosin (H&E) stains were performed using standard protocols. IHC for CSF2RA (Colony Stimulating Factor 2 Receptor Subunit Alpha) was undertaken on 3 μm sections using a Leica Bond-max automated platform (Leica Biosystems). In brief, antigen retrieval was performed to unmask the epitope (Heat Induced Epitope Retrieval (HIER), Bond-max protocol F), endogenous activity was blocked with peroxidase using a Bond polymer refine kit (cat # DS9800), then incubation was undertaken with a primary rabbit polyclonal CSF2RA (GM-CSF receptor alpha) antibody for 1 h (Origene TA323990S, 1:100 dilution, HIER1 for 30 min). A post-primary antibody was applied to the sections (Bond polymer refine kit) and horseradish peroxidase (HRP) labeled polymer, followed by 3, 3-diaminobenzidine (DAB) chromogen solution. Sections were counterstained with hematoxylin, washed, dehydrated, cleared in two xylene changes and mounted for light microscopy. Images were taken on an Aperio CS2 Scanner (Leica Biosystems) at 40x objective. Analysis was performed with QuPath (v.0.2.3) (https://qupath.github.io) software.

### Tissue and placental expression of key genes

General expression of *AACAD3L* and *CSF2RA* in adult tissues was performed using GTEx Consensus bulk RNA-seq consensus data. Single-cell RNA-seq expression of these genes in first trimester placenta was visualized using the CELLxGENE[44] repository to access data by Vento-Tormo et al.[42] (https://maternal-fetal-interface.cellgeni.sanger.ac.uk/) (CC-BY-4.0). Single-cell RNA-seq expression of these genes in term and pre-term placenta was visualized using data by Pique-Regi et al.[43] (http://placenta.grid.wayne.edu/) (CC-BY-4,0). Consensus tissue datasets were obtained from the Human Protein Atlas for *AADACL3* expression (https://www.proteinatlas.org/ENSG00000188984-AADACL3/tissue) and *CSF2RA* expression (https://www.proteinatlas.org/ENSG00000198223-CSF2RA/tissue) (Data accessed 10/17/23; Protein Atlas version 23.0) (Uhlén et al.)[65] (CC-BY-4.0)

### Statistics and reproducibility

Statistical analysis for quantitative variability in relative expression between karyotypes was undertaken using one-way ANOVA (Kruskal-Wallis) test in GraphPad Prism and presented using GraphPad (version 9.5.1 for Windows, GraphPad Software, San Diego, CA, USA, (www.graphpad.com). Statistical significance associated with differentially expressed genes were carried out in R, as defined above[70,71]. A p-value of <0.05 was considered significant, following adjustment for multiple comparisons where indicated. The sample sizes are defined in detail in the relevant section of the methods, in figure legends and in supplementary data tables. In general, $n = 4$ samples were used for each tissue/karyotype group and $n = 20$ for the multi-tissue analysis. Placental samples ranged from $n = 6$ to $n = 18$, depending on the hypothesis being addressed. All samples were independent and groups balanced with controls, except for the use of a bisected pancreas sample at 13wpc, which showed biological variability and which we justify in the methods. As several samples in the multi-tissue analysis came from the same fetus, we undertook an additional linear mixed methods analysis without seeing any marked differences (Supplementary Data 42).

### Reporting summary

Further information on research design is available in the Nature Portfolio Reporting Summary linked to this article.

### Data availability

Bulk RNA-sequencing data are deposited in ArrayExpress/Biostudies (accession number E-MTAB-13673). Supplementary Data (1-49) and source data (Fig. 3b to Fig. 9h; Supplementary Figs. 1b to 12b) have been deposited in Open Science Framework (https://doi.org/10.17605/OSF.IO/U3485)[79]. All other data are available from the corresponding author as applicable, on reasonable request.

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

## Acknowledgements
This research was funded in whole, or in part, by the Wellcome Trust (J.C.A. 209328/Z/17/Z; S.M. McB 216362/Z/19/Z). For the purpose of Open Access, the author has applied a CC BY public copyright license to any Author Accepted Manuscript version arising from this submission. Human fetal material was provided by the Joint MRC/Wellcome Trust (Grant MR/R006237/1, MR/X008304/1 and 226202/Z/22/Z) Human Developmental Biology Resource (http://www.hdbr.org). All research at UCL Great Ormond Street Institute of Child Health is made possible by the NIHR Great Ormond Street Hospital Biomedical Research Centre. The views expressed are those of the authors and not necessarily those of the National Health Service, National Institute for Health Research, or Department of Health. The Genotype-Tissue Expression (GTEx) Project was supported by the Common Fund of the Office of the Director of the National Institutes of Health, and by NCI, NHGRI, NHLBI, NIDA, NIMH, and NINDS. The data used for the analyses described in this manuscript were obtained from the GTEx Portal on 10/10/23.

## Author contributions
J.P.S. and J.C.A. conceptualized the study. J.P.S., I.D.V., F.B., S.M.B., T.B., O.O., D.L., N.D., G.M.K., K.M., L.N., A.R.M., M.I., N.J.S., G.E.M., B.C., N.S., G.S.C. and J.C.A. undertook data curation. J.P.S. and S.M.B. undertook formal data analysis. J.C.A. was involved in funding acquisition. J.C.A. and N.S. oversaw project administration and supervision. I.D.V. undertook validation of bioinformatic platforms and pipelines. J.P.S. and F.B. were responsible for data visualization. J.P.S. and J.C.A. wrote the original draft with input from GSC. All authors were involved in reviewing and editing the final manuscript. J.P.S. and J.C.A. had full access to all data in the study and had final responsibility for the decision to submit for publication.

## Competing interests
The authors declare no competing interests.
