## [Transparent Peer Review file · Communications Biology]

The transcriptomic landscape of monosomy X (45,X) during early human fetal and placental development

Corresponding Author: Professor John Achermann

This manuscript has been previously reviewed at another journal. This document only contains information relating to versions considered at Communications Biology.

Version 0:

Reviewer comments:

Reviewer #1

(Remarks to the Author)

Suntharalingham et al. studied early fetal and placental development in 45,X Turner syndrome. They studied 3-4 fetuses with 45,X (one turned out to be low level mosaic) and a similar number of male and female controls, matched for age. They describe transcription changes in different organs, pancreas, liver, kidney and skin, similar to what has been described in adult tissue from TS, albeit not all these tissues have been studied in adults. They describe downregulation of primarily PAR1 and X-Y homologous genes in these different organs, but also tissue differences. They also study placental tissue. Interestingly, they find that OVCH1-AS1, presumably a non-coding RNA, to be considerably upregulated. It is currently not clear what OVCH1-AS1 does – could it be involved in survival?

1. The authors have sampled several interesting tissues, but I wonder why they did not sample the heart and brain, which are very essential in TS? I see that they analyzed them in the “mixed group”.
2. The samples come from Medical Research Council (MRC)/Wellcome Trust-funded Human Developmental Biology Resource (HDBR) – how can the authors be sure that the material has been kept in conditions whereafter meaningful amounts of RNA can be extracted?
3. Since the authors did not themselves assemble the fetuses, I assume that the reason for the low sample size is that MRC do not have any more relevant fetal samples?
4. It is very important to know if the fetuses are from spontaneous abortions or from legal abortions. The authors need to present this information. Given that most 45,X TS fetuses are spontaneously aborted, some of the changes presented here could thus be associated with spontaneous abortion. However, if the fetuses could have been future “survivors”, some of the observed transcription changes could be seen as changes that are necessary for a TS fetus to survive.
5. I’m not sure I understand why the authors in one instance choose to sample a 45,X pancreas twice, as a biological replicate? This is not done in any other cases of the tissue samples and would therefore introduce a bias towards this one sample. This is also done for the placentas.
6. The authors find lower expression of the PAR1 gene, SLC25A6, in multiple tissues, which has also recently been found by others, and here linked to length of the QT interval - PMID: 37495650. Could the expression of SLC26A6 here be linked to part of the cardiovascular phenotype?
7. Figure 3 nicely summarizes the results concerning expression of PAR1 genes. However, it would be interesting to also add genes considered to be X-Y homologues, as they would be expected to follow the same pattern of universal downregulation.
8. It is interesting to note that most PAR1 genes in the comparison with 46,XX do not show the expected downregulation of about -1, but in fact closer to -0.5. This suggests that some kind of compensatory mechanism must be at play? Would the authors have any suggestion or data that can explain this?
9. In Figure 4 a and b it could be a good idea to add the expression value from the Y homologue in the 45,X vs 46,XY comparison, in order to illustrate the likely discrepant resultant expression of for example KDM5C.
10. Line 191 – it should be KDM5C and not KDM5D.
11. The data concerning CSF2RA are very intriguing and resemble old data from Urbach and Benvenisty. A mechanistic study with for example placental organoids would be of great interest to look more into this.

12. In figure 8, the authors describe inflammatory changes with an irregular villus border and they link this to changes in CSF2RA expression. However, did the authors look for other signs of perturbed immune function in their data? Could other elements of the immune system be dysfunctional?
13. It is also very interesting that there does not seem to be evidence of placental mosaicism, but I wonder you can be completely sure of that? Could sampling in other places of the placenta present different results? In fact the authors do not explain where and how they sampled the placentas. This bit of information should be added.
14. The statistical analysis seems appropriate.

Reviewer #2

(Remarks to the Author)

In this study, Suntharalingham et al. have evaluated the transcriptional profile of various clinically-relevant tissues from late first trimester to early second trimester fetuses with monosomy X (45,X) compared to gestational-age and tissue-matched 46,XX and 46,XY controls to study differences in early fetal development associated with monosomy X and identify potential developmental origins and underlying mechanisms of the various phenotypic presentations of monosomy X/Turner syndrome.

This is a unique study and is the first to assess the transcriptome of these diverse tissues from 46,XX, 46,XY, and 45,X fetuses. Some results replicate findings that have been reported in previous studies of monosomy X from other tissues and/or stages of life, while other findings are original and add to our understanding of the monosomy X transcriptome during development – both are potentially important to provide insights into the underlying pathogenic mechanisms of this aneuploidy. These results should all be interpreted with caution however, as the small sample sizes (differential gene expression analyses based on groups of maximum N=4 for other fetal tissues and N=6 for placenta) are a limitation to drawing strong conclusions from the results of this study. This is potentially understandable given the source of the samples used for this study, and while results are novel and interesting to the field, the report could merit from being framed as an exploratory or pilot study. By making the RNA-seq data generated in this study publicly accessible, the authors have provided an excellent resource for gene expression studies in these unique and rare tissue samples for future studies. They have also provided extensive supplemental data to be transparent about the results of the study.

Specific comments:

1) The small sample sizes used in the various differential gene expression comparisons are a potential limitation to drawing strong conclusions in this study. With only 4 individuals per group for the fetal tissues comparisons and 6 for placental comparisons, any stochastic differences in inter-individual gene expression cannot be well controlled for and may influence results. The authors somewhat address this indirectly through the Principal Components Analysis (PCA), showing that top drivers of variation in gene expression among all tissues is tissue type, and within most individual tissues is karyotype, and also by looking for expression differences that are in common across tissues and across the 46,XX and 46,XY comparisons to try to identify replicated differences. Despite this, further evaluation of drivers of gene expression variation were not assessed, or are not mentioned in the manuscript, beyond Principal Components 1-2. Particularly within the “all tissue” PCA, were other factors such as gestational age or individual (e.g. each genetically unique individual – because multiple tissues were contributed from the same individual, particularly for 45,X cases) assessed for association with further PCs? This could help emphasize whether differences by karyotype outweigh any potential inter-individual differences, and strengthen the findings despite the small sample sizes.

In addition to further support from the PCA, the authors could consider framing this study as more of an exploratory or pilot study given the small sample sizes and being more upfront about this throughout the manuscript – e.g. reminding readers of sample sizes for each of the comparisons by providing N's within the text of results section and in the figures.

2) In Lines 106-108, the authors mention how factors such as medication, inflammation, diet, etc. may confound transcriptomic profiles of adult tissues, however they do not address potential confounders of fetal or placental tissue within their own study design in the manuscript. Biological factors such as maternal health conditions or exposures can and have been shown to influence transcriptional profiles of fetal tissues including placenta (e.g. maternal cigarette smoking, maternal diabetes); other yet-detected sub-chromosomal genetic anomalies that may influence early development were not screened for and may also be possible confounders in this study. Additionally, technical factors such as sample processing time may also influence transcriptional profiles. Given that these tissues were acquired from a biobank, such information may not be readily available, but should at least be addressed as a potential confounder or limitation of the study, particularly given the small sample sizes.

3) Given that a unique contribution of this study is the use of these various disease-relevant tissues, while reporting on differentially expressed genes between the 45,X and controls that are in common between each of the tissues is of interest, the authors could comment on the differentially expressed genes that may be unique within each given tissue between the Turner and control samples. Particularly if these genes are key for the function of these tissues, or are uniquely expressed in these tissues, this could be worthwhile to highlight as potential insight into the different Turner syndrome associated phenotypes observed within these tissues. This is somewhat shown in Supplementary Figures 4 and 7 but could be addressed more definitively within the manuscript.

4) In lines 275-285, the authors present their findings of a lack of mosaicism for 46,XX cells in the 45,X placentas (N=6), concluding that this mosaicism is likely not common. This is an interesting finding, and they discuss how this potentially

challenges previous hypotheses that cases with 45,X that survive to term are mosaics for other viable karyotypes. However, this conclusion should be put in context of a) the small number of placental samples tested b) the timing of sample collection – given that ~98% of recognized 45,X pregnancies do not survive to term, based on the timing of 45,X losses, is it possible that some of these fetuses may have been spontaneously lost later in pregnancy and represent part of this larger group of pregnancies that are not “rescued”?; and c) any potential ascertainment bias – because samples included in this study were selected on the basis of having a 45,X karyotype, could this group of placentas simply just be enriched for cases that are non-mosaic/very low-level 45,X karyotypes? Further explanation of these factors would be helpful to support the conclusion being drawn.

5) In line 339, the authors report on results of pathway enrichment for genes with lower expression in 45,X placenta vs 46,XY placenta, highlighting enrichment of processes related to immune function. They do not however comment on any pathway enrichment in 45,X vs 46,XX placentas. Was this performed? If so, this should be mentioned and a comment on whether the immune-related pathway enrichment is consistent or not. If it was not performed, a justification for this should be given.

6) While not a study of direct fetal tissues such as this one, a pilot study by Massingham et al. (2014) also assessed transcriptional differences in 45,X vs 46,XX fetuses based on cell-free RNA in amniotic fluid (AF). Cell-free nucleic acids in AF are primarily fetal in origin and expected to arise from fetal tissues in direct contact with AF such as skin, therefore given that this is one of the few other studies assessing transcriptional differences in Turner syndrome in prenatal development, a mention and comparison to the results of this study would be a helpful addition to the discussion. In particular, the authors have some similar findings as in this study, eg. SHOX not differentially expressed, low expression of LDLR.

Massingham, L. J., Johnson, K. L., Scholl, T. M., Slonim, D. K., Wick, H. C., & Bianchi, D. W. (2014). Amniotic fluid RNA gene expression profiling provides insights into the phenotype of Turner syndrome. *Human genetics*, 133(9), 1075–1082.

Reviewer #3

(Remarks to the Author)

Summary & overall impression

This interesting work profiles the transcriptomes of 45,XO prenatal tissue samples using mRNA sequencing to probe the biological consequences of having only one X early in developmental in light of the high rate of spontaneous abortion seen in monosomy X conceptuses and specifically evaluates the hypotheses of reduced PAR/escape gene expression driving monosomy X phenotypes. This study is important to publish as it provides information on a timepoint that is difficult to study, particularly using a multi-tissue approach, however, the analytical methods would benefit from revision as the methods implemented are not the most statistically powerful for this study design.

Comments

- Major 1: I am not convinced that the statistical approach taken (linear modelling) is valid for this study design. Due to the sample distribution presented in Supp figure 1, when comparing between tissue types the authors I think have a mixture of some genetically identical samples (with same donor contributing multiple tissues), alongside samples that are only donors for one tissue type. In this case, a linear mixed effects model controlling for individual as a random effect would be a more appropriate choice for all of the linear models presented. In addition, rather than stratifying comparisons between 45,X and the 46,XX or 46,XY samples by tissue, it would also be more statistically powerful to run one model including all samples + tissue types, with an interaction term for tissue*karyotype. This is under-appreciated, but stratifying data by sex, etc, reduces power to discover true positives. At the very least, a mixed effects analysis could be included as a supplementary sensitivity analysis. For a further discussion of this topic, see Chin & Christians 2015, PMID26283072, <https://pubmed.ncbi.nlm.nih.gov/26283072/>

- Major 2: Genes with lower expression in monosomy X (line 159) and genes with higher expression in monosomy X across tissues (line 220) – intersecting adjusted p-value lists of differentially expressed genes is not a sufficiently powerful way to conduct this type of comparative analysis and leads to overestimation of false negatives or lack of overlap (see reference suggested above, Chin & Christians 2015). A stronger alternative would be to consider whether the effect sizes of DEGs at nominal significance are correlated between tissues (i.e. logFC correlations between tissue types) and to select those that are most strongly correlated to determine a consensus list of genes that are affected by karyotype.

- Sampling methods of the clonally-developed placenta should be described as thoroughly as possible in the Methods section, though I understand the difficulty associated with early-gestation placental sampling.

- What were the results of RNA sample QC with TapeStation prior to sequencing (Methods line 546)? It is important to report the metrics assessed (RIN, DV200, etc) as well as the average results of these assessments, as I imagine tissue may be confounded with RNA quality, and that context is important for manuscript interpretation.

- As noted by the authors, in the introduction (line 87, reference 25), San Roman et al. recently identified X-linked genes that were proposed to drive somatic sex differences, the relative expression of the genes ID'd by San Roman are also evaluated explicitly in the Results section and reported on in Supp Data 30. However, more recent work has been published by the same group defining autosomal genes that are moderated by the impact of an inactive X or Y chromosome. The loss of either of these would lead to monosomy X and it could be presumed for similar reasons that many of these genes may be affected in 45,XO samples. As such it would be valuable to extend the analysis in lines 181-192 to include the most recent San Roman et al. 2024 findings and evaluate the effect of karyotype on genes, or use this list to contextualize the findings presented in lines 194-212. San Roman et al. 2024: PMID38190107 <https://www.ncbi.nlm.nih.gov/pmc/articles/PMC10794785/>

- Supplementary Figure 1 is slightly confusing, in addition to being biological replicates if two half circles are present in the same tissue type, they also appear to be technical replicates? i.e. do two half-circles (purple) in the pancreas category

indicate that the same pancreas sample was transcriptome sequenced 2x? If so, were these technical replicates removed prior to linear modelling? Were they used for any other quality assessments? This should be clarified and confirmed in the results section of the manuscript around line 149.

- Additionally, was gestational age (or any other covariate) accounted for in the linear model presented in Figure 2?
- I appreciate that placenta was not included in the DEG analyses presented in Figures 2-5 as it is analysed separately in the later section, however the manuscript would benefit from addressing the motivation for analyzing placenta separately earlier, i.e., in the introduction. Additionally, a potentially useful paper to add to your evidence for placental intolerance of monosomy X would be Ahern et al. 2022: <https://pubmed.ncbi.nlm.nih.gov/36161909/>
- It may be more valuable to present supplementary Figure 5 sorted by karyotype on the X axis and colored by tissue.
- The logic for only analysing chromosome 12 genes related to the lncRNA expression was not clear in the paper, I assume it was simply due to a higher probability of detecting cis effects? This could be clarified.

Version 1:

Reviewer comments:

Reviewer #1

(Remarks to the Author)

No additional comments. The authors have done a very good job in responding to my comments.

Reviewer #2

(Remarks to the Author)

I thank the authors for their detailed responses and modifications to the original manuscript to address my and other reviewers' comments. The additional details and analyses included in response to reviewer comments improve clarity and strengthen the manuscript. Based on the revised manuscript and responses provided, I believe the authors have sufficiently addressed my concerns.

Reviewer #3

(Remarks to the Author)

Thank you for revising your work according to our feedback. In particular,

- Thank you for including the linear mixed effects model and results. Although the results are largely consistent with your previous analysis in this dataset, it is valuable and adds confidence in your other results to see the mixed effects model, given that it is the most appropriate test for this type of analysis.
- Fewer significant results after running an interaction test are expected, as likely you are losing the false positives. I would have appreciated seeing this addressed in at least one sentence in the manuscript.
- The correlation plots largely show what the authors had earlier hypothesized, and the differences in strength of correlation in the various tissue comparisons is interesting. Clearly some tissues (as expected) are more consistent with each other than with others.

RESPONSES TO REVIEWERS

COMMSBIO-24-0843-T

The transcriptomic landscape of monosomy X (45,X) during early human fetal and placental development

Suntharalingham et al.

Reviewers' comments:

Reviewer #1 (Remarks to the Author):

Suntharalingham et al. studied early fetal and placental development in 45,X Turner syndrome. They studied 3-4 fetuses with 45,X (one turned out to be low level mosaic) and a similar number of male and female controls, matched for age. They describe transcription changes in different organs, pancreas, liver, kidney and skin, similar to what has been described in adult tissue from TS, albeit not all these tissues have been studied in adults. They describe downregulation of primarily PAR1 and X-Y homologous genes in these different organs, but also tissue differences. They also study placental tissue. In interestingly, they find that *OVCH1-AS1*, presumably a non-coding RNA, to be considerably upregulated. It is currently not clear what *OVCH1-AS1* does – could it be involved in survival?

We thank the reviewer for their interest in this manuscript. The role of *OVCH1-AS1* is intriguing, and could be followed up in future work. It could potentially be involved in “survival” pathways; as *OVCH1-AS1* is consistently higher in monosomy X, one could hypothesize that it is a negative regulator of survival. The linked gene, *OVCH1*, has been proposed to be involved in regulating apoptosis of ova and sperm as a potential quality control mechanism. If increased *OVCH1-AS1* reduced levels of *OVCH1* RNA that would have been a potentially interesting link. However, we did not see any differences in *OVCH1* expression by RNA-seq and independent qRT-PCR, so we did not want to hypothesize or speculate further at this stage.

1. The authors have sampled several interesting tissues, but I wonder why they did not sample the heart and brain, which are very essential in TS? I see that they analyzed them in the “mixed group”.

Our plan was to obtain 4 different samples for each main tissue group. Unfortunately, we could not achieve this because of sample availability for heart and brain, so decided to include single samples of these tissues in the “mixed” group. This actually was a useful approach as using tissues from diverse embryological origins potentially reduces (or smooths) tissue-specific effects, so that more specific monosomy X-related effects could be seen. This especially provided further validation for the main findings and novelty, such as *OVCH1-AS1*.

We have added the following:

Lines 585-587: “Each organ in the mixed tissue group (brain, spleen, lung, and heart) was collected at a different age stage within the timeframe (11 to 15 wpc), to give a further perspective on other key tissues (Fig. 1a).”

2. The samples come from Medical Research Council (MRC)/Wellcome Trust-funded Human Developmental Biology Resource (HDBR) – how can the authors be sure that the material has been kept in conditions whereafter meaningful amounts of RNA can be extracted?

We can address this with general and specific responses:

a) The HDBR was set up in 1999 and it is a well-established biobank that provides ethically-approved access to human embryonic and fetal materials (www.hdbr.org). There have been more than 200 publications, including several highly-cited studies, and our group have published more than 20 papers in collaboration with HDBR over this time.

b) There are well-defined standard operating procedures in place. Samples are rinsed and stored in L15 media on ice packs, then transported rapidly to the lab for processing by experienced research staff. Dissected tissues are then frozen at -70 degrees or placed in appropriate media/formalin.

From available data, the mean maximum time to dissection was: 45,X samples, 90 minutes; 46,XX samples, 114 minutes; 46,XY samples, 117 minutes. These times are typical of targets for the Biobank.

c) There were no issues with the quantity of tissue available, as the manufacturer's recommendation for RNA extract is 10-30mg tissue.

d) All RNA samples underwent QC using Nanodrop and Tapestation. The RIN (RNA Integrity number) was similar for 45,X samples and controls, and these values are now shown in Supplementary Figure 1b and below. Placenta RINs are typically lower because of RNase activity but good quality libraries were obtained. There were no differences between RIN values in the different groups (Kruskal-Wallis test). Data are provided in supporting data.

b RNA integrity number (RIN) values for samples used in this study. Groups were compared using Kruskal-Wallis analysis. ns, not significant.

The following changes have been made in the text:

Lines 561-563: Samples were obtained following termination, and initially stored in Leibovitz's L15 media (Gibco/Thermo Fisher Scientific, Waltham, MA, USA) on ice (mean < 2h) before being frozen (-70°C) or transferred to formalin.

Lines 615-616: RNA integrity numbers (RINs) are shown in Supplementary Fig. 1.

3. Since the authors did not themselves assemble the fetuses, I assume that the reason for the low sample size is that MRC do not have any more relevant fetal samples?

That is correct; the availability of monosomy X samples following termination of pregnancy is low. This study accessed samples over a 2-3 year period and we were able to obtain tissues from 6 fetuses during this time. These challenges add to the unique aspect of our study, as noted by reviewers, and the importance of making these data available for the wider community, and ultimately for potential benefit for girls and women with Turner syndrome.

4. It is very important to know if the fetuses are from spontaneous abortions or from legal abortions. The authors need to present this information. Given that most 45,X TS fetuses are spontaneously aborted, some of the changes presented here could thus be associated with spontaneous abortion. However, if the fetuses could have been future "survivors", some of the observed transcription changes could be seen as changes that are necessary for a TS fetus to survive.

These samples were all from legal terminations, that were counselled and consented as discussed in the methods. There was no tissue included from spontaneous loss of pregnancy, as this would likely have occurred in an “out-of-clinic” setting, and tissues would not have been preserved using standard operating procedures as we now state above. Further information about the consent process and donation of materials is provided on the HDBR website (www.hdbr.org).

We now clarify:

Line 561: **“Samples were obtained following elective termination,.....”**

The reviewer raises the important point about whether these pregnancies would have survived to term or not. Most pregnancy losses associated with monosomy occur in the first trimester, to our knowledge. However, later spontaneous loss of pregnancies in the second trimester can occur. Our samples were between 11-15 weeks post conception (13-17 weeks gestation), so there is a chance loss of pregnancy could have occurred still. If there were significant fetal anomalies resulting in early pregnancy loss (such as hydrops fetalis), then this would have been obvious and the transcriptomic profile may have been different. We suspect that what we are assessing here is the “fetal” transcriptome of monosomy X pregnancies that may reach term, but we do not know with certainty.

As far as the placenta is concerned, we did see general changes such as mild edema and the villous brush border defects, together with the previously postulated reduction in *CSF2RA* and other genes. If we assume that these pregnancies would have reached term, then we think it is less likely that we are seeing specific rescue mechanisms, but rather that the disruptive mechanisms in the samples studied are less severe than in most monosomy X pregnancies, which might be lost earlier. We could only really address this hypothesis by also comparing the transcriptomic (and genomic) profiles and paired histology of placental tissue from early embryos that undergo spontaneous loss (which would likely be extremely poor quality RNA) with monosomy X placentae at term. Obviously, this would be a extremely difficult study to undertake, for many reasons

Although we eluded to this Reviewer’s point previously, we have re-written the end of the discussion to address this further:

Lines 523-527 (Discussion): **“Furthermore, as we studied placental material at 11-15wpc, we do not know if the samples we analysed harbored any specific features that protected against pregnancy loss at an earlier stage (first trimester), or whether the histological and transcriptomic changes seen could still have been associated with pregnancy loss at a later gestational age.”**

5. I’m not sure I understand why the authors in one instance choose to sample a 45,X pancreas twice, as a biological replicate? This is not done in any other cases of the tissue samples and would therefore introduce a bias towards this one sample. This is also done for the placentas.

Pancreas

We had to bisect one pancreas along the long axis and use the two halves, as we were only able to obtain three monosomy X pancreas samples for this study within the time frame, and we really wanted four samples to increase the power, and also to have balanced groups for direct comparison with other tissue. If we had used three

samples in the pancreas group and four in all others, it would not have been appropriate to compare datasets based on a fixed adjusted p-value, as this would have been affected by sample size.

By bisecting a pancreas sample on the long axis, we expected to get biological variability as it is not a homogenous structure and – at that age – contains developing ducts and regions of endocrine cell clusters. We felt the biological variability of these two parts would be much greater than the genetic contribution of the individual, especially at a transcriptomic (rather than genomic) level. This was confirmed by looking at the PCA plot (below). The single bisected pancreas (45,X, 13 wpc) obviously clusters with the other pancreas samples, but does show biological variability between the two parts (arrows), with similar variability to that seen in the 46,XX and 46,XY pancreas clusters. It can be clearly seen that these two 13wpc samples are not superimposed in the principal component analysis. Thus, we felt strongly that it was justified and best to include this bisected single sample in the downstream analysis, but obviously to be transparent about it.

We have clarified this approach further by making the following changes:

Lines 579-585 (Methods): “Because of limited tissue availability for the 45,X pancreas group, a 13 wpc sample was bisected sagittally from the head of the pancreas to the tail, so that RNA could be independently from two halves of the same biological sample. These two parts showed a degree of biological variability similar to other pancreas samples (Fig. 1c), suggesting that the shared origin of the samples was not a major driver of the transcriptomic profile, and that including both samples in the analysis to obtain n=4 was justified.”

Lines 531-532 (discussion limitations): We have added: “...and one 45,X pancreas sample (13 wpc) was bisected in order to have four samples in this group.”

Figure 1c legend – We already state (lines 1091-1092): “One 13 week post conception (wpc) pancreas was bisected longitudinally and both parts processed independently” We have now added (lines 1093-1094): “(Note: the two 45,X samples at 13 wpc showed biological variability).”

We also address aspects of this in our response to reviewer 3.

Placenta

The main placental analysis was undertaken with six samples, one from each placenta, and six matched 46,XX and 46,XY controls. There were no multiple samples from the same placenta used for the main analysis.

Multiple samples from the same placenta (45,X) were only used to address the hypothesis of confined placental mosaicism. Here, multiple samples are essential to address this hypothesis.

6. The authors find lower expression of the PAR1 gene, SLC25A6, in multiple tissues, which has also recently been found by others, and here linked to length of the QT interval - PMID: 37495650. Could the expression of SLC26A6 here be linked to part of the cardiovascular phenotype?

Thank you for highlighting this recent publication. We have now mentioned this in the introduction, where PAR genes may have specific effects such as SHOX/short stature, and **added the reference**. As only a single heart sample was included in the “mixed” group, we were unable to address this in any more detail in a tissue-specific manner.

Line 81: “.....**or cardiac QTc interval duration**”

7. Figure 3 nicely summarizes the results concerning expression of PAR1 genes. However, it would be interesting to also add genes considered to be X-Y homologs, as they would be expected to follow the same pattern of universal downregulation.

Thank you for raising this interesting point. Several X-Y homologs were included in Fig. 4a and 4b linked to the Viuff et al and San Roman et al (2023) data, and of course these are in the extensive Supplementary Data we provided. However, we agree this is an important point and we have now addressed this overview more systematically.

We focussed on the core “ancestral” non-PAR X-Y homologs defined by Godfrey et al (Genome Research 2020 30:860-873; Fig 1). We have generated data plots for individual tissue analysis for these key genes in the same style as the Viuff and San Roman data, and present these data as additional panels in the same figure (New Fig. 4d and 4e). We present the X chromosome gene directly above the corresponding Y chromosome gene, so that relative differences can be easily seen. This additional analysis provides important additional information about key X-Y homologs that show altered expression in monosomy 45,X tissues, and which may contribute to the mechanisms of Turner syndrome.

For balance and clarity, we have now moved the cartoon of some key genes in Viuff/San Roman to Fig. 4a. We have also labelled new Fig. 4b and 4b with the title of the datasets they are derived from (“Viuff et al. 2023” and “San Roman et al. 2023”) and highlighted in the legend that these represent proposed “core” genes in monosomy X and core “sex differences” genes, respectively. We have moved the previous analysis of genes lower in monosomy/pathways (old Fig 4d-f) to a new Fig. 5 a-c, with corresponding legend, and this is a standalone analysis that follows on from the X-Y homolog question. Subsequent figure labelling has changed in the manuscript. We also include the data and analysis in new supplementary data 31-33.

The new Fig 4 legend is:

Figure 4. Additional key genes proposed to have lower expression in monosomy X, including X-Y homolog pairs. **a** Schematic showing the genomic location of key genes of interest on the X chromosome (see Fig. 4b, c). PAR genes are not shown. **b** Differential expression in our datasets of selected "core" X chromosome genes that have been linked to monosomy X, including those proposed to show X chromosome dosage (from Viuff et al, 2023). Individual mean data points for each tissue group are shown, as indicated in the legend. The bars represent the mean of these different tissue groups with standard error of the mean shown. Log2 fold change (FC)=-1.0 represents half the expression in 45,X samples (i.e., haploinsufficiency), whereas log2FC=0 represents similar expression in 45,X samples and 46,XX and 46,XY controls (n=4 for each karyotype in each tissue group). Note *XIST* and *TSIX* show strong differential expression in all datasets but are omitted from the graphic. **c** Differential expression in our datasets of selected X chromosome genes linked to "sex differences" (from San Roman et al, 2023) (25). Data are presented as described above. Note the different y-axis scale compared to Fig. 4b. **d** Differential expression in our datasets of key "ancestral X-Y homolog" gene pairs (from Godfrey et al., 2020). This panel shows the X chromosome gene of the pair, with the corresponding Y chromosome homolog gene below it (Fig. 4e). Data are presented as described above. **e** Differential expression in

our datasets of key “ancestral X-Y homolog” gene pairs (from Godfrey et al., 2020). This panel shows the Y chromosome gene of the pair, with the corresponding X chromosome homolog gene above it (Fig. 4d).

The new analysis of ancestral X-Y gene pairs provides several insights:

a) There are no Y gene differences between 45,X and 46,XX (Fig. 4e, upper panel) as expected, as no Y chromosome is present (and the presence of a low level Y line in isolated monosomy X samples does not have a clear impact on the dataset).

b) There are limited X gene differences between 45,X and 46,XY samples (Fig. 4d, lower panel), with variation around log₂FC “0”, as these samples all have a single X chromosome that does not undergo X inactivation.

c) Analysis of 45,X versus 46,XX (Fig. 4d upper panel) and 45,X and 46,XY (Fig. 4e lower panel), clearly shows several X-Y homologs that are consistently lower in monosomy X. These include *EIF1AX/EIF1AY*, *ZFX/ZFY*, *DDX3X/DDX3Y*, *KDM6A/UTY*, *KDM5C/KDM5D*, and *RSP4X/RSP4Y1*. These genes could contribute mechanistically to the features of monosomy X.

Taken together, we feel this additional analysis provides important additional information and thank the reviewer for raising this.

We have therefore changed several parts of the text to include this:

Lines 41-42 (Abstract): “.....we also found reduced expression of several key genes escaping X inactivation (e.g., *KDM5C* and *KDM6A*), several ancestral X-Y gene pairs, and potentially clinically important transcripts...”

Lines 192-202 (Results):

We replaced:

“Although similar expression of these two genes was seen between 45,X and 46,XY tissues, relative haploinsufficiency of demethylase activity is still likely to occur in monosomy X compared to 46,XY tissues as 46,XY tissues have compensation by their Y chromosome homologs, *KDM5D* (also known as *JARID1D/SMCY*) and *UTY* (*KDM6C*), respectively. Thus, histone methylation status may be altered in 45,X and could influence developmental processes”.

With:

“As *KDM5C* and *KDM6A* are X-chromosome genes with Y-chromosome homologs (*KDM5D* and *UTY*, respectively), we extended our analysis to look at differential expression of homologous, ancestral X-Y gene pairs (as defined by Godfrey et al., 2020) (Fig. 4 d, e and Supplementary Data 31-33. Most X-Y gene pairs showed a decrease in both 45,X versus 46,XX and in 45,X versus 46,XY datasets, with the most consistent and marked lower expression in monosomy X for *EIF1AX/EIF1AY*, *ZFX/ZFY*, *DDX3X/DDX3Y*, *KDM6A/UTY* (also known as *KDM6C*), *KDM5C/KDM5D* (also known as *JARID1D/SMCY*), and *RSP4X/RSP4Y1*. Many of these genes play a role in transcription, translation and histone methylation status, which may be altered in 45,X and could influence developmental processes.”

Lines 403-407 (Discussion): “By analysing ancestral X-Y gene pairs (Godfrey et al, 2020), we showed consistently lower expression in monosomy X for *EIF1AX/EIF1AY*, *ZFX/ZFY*, *DDX3X/DDX3Y*, *KDM6A/UTY*, *KDM5C/KDM5D* and *RSP4X/RSP4Y1*. Although these genes can influence transcription and translation, the role of histone demethylases (e.g., *KDM5C/KDM5D*, *KDM6A/UTY*) may be particularly important.”

Lines 410-412 (Discussion): “Here, we report, for the first time, core groups of PAR and XCI genes, and ancestral X-Y gene pairs that show lower expression in key biologically-relevant fetal monosomy X tissues.....”

To accommodate changes in Fig. 4, we have moved old Fig. 4 panels d-f into a new Fig. 5, as this also thematically more cohesive.

New Figure 5:

8. It is interesting to note that most PAR1 genes in the comparison with 46,XX do not show the expected downregulation of about -1, but in fact closer to -0.5. This suggests that some kind of compensatory mechanism must be at play? Would the authors have any suggestion or data that can explain this?

Thank you for raising this point, which we have considered. As we mention, some PAR1 genes do not seem to have a difference and these are often expressed at very low transcript level or have tissue specific expression (e.g. *SHOX*, *ASMT*). We agree that some of the other more consistently lower genes do not show the log2FC -1.0 level that might be expected for haploinsufficiency. This trend is seen to some degree also in the data of Viuff et al., 2023, Fig 2. To us, this is more apparent in the 45,X versus 46,XX comparison, rather than the 45,X versus 46,XY comparison. We have reverted to look at raw data for normalized counts in case there was an effect of log2 conversion, but the counts show a similar pattern. Potentially there could be a compensatory mechanism in comparison to 46,XX. There did not seem to be any outlier tissues that were consistently different. Whilst interesting, we did not feel we wanted to enter lengthy discussion about this, especially given the space constraints for the manuscript, but this might be worth pursuing in other datasets.

9. In Figure 4 a and b it could be a good idea to add the expression value from the Y homolog in the 45,X vs 46,XY comparison, in order to illustrate the likely discrepant resultant expression of for example KDM5C.

Thank you. We have now addressed this point by bringing in extensive additional data on X-Y homologs, to the figure panel (see response to point 7 above) and supplementary data. (In Fig 4a (was Fig. 4c) we have also corrected the KDM6A to UTY for 46,XY).

10. Line 191 – it should be KDM5C and not KDM5D.

KDM5D is correct as line 191 gives the Y homolog of the X gene KDM5C in line 187. To make it clearer, we have put aliases for genes in parentheses, and said “also known as”. For examples, “KDM5D (also known as JARID1D/SMCY)”.

11. The data concerning CSF2RA are very intriguing and resemble old data from Urbach and Benvenisty. A mechanistic study with for example placental organoids would be of great interest to look more into this.

Thank you for this comment. We cite this paper (now ref 7) in the introduction and specifically in relation to PAR genes and CSF2RA in several places the discussion (line 484). We say that: “Several studies have proposed that PAR genes such as *CSF2RA* are implicated, but direct data are limited^{2,6}” (line 483-484), and also discuss this paper in terms of it being a candidate gene for early lethality (line 503) and effects on fold expression (lines 504-506).

In light of the comments, we have made the following changes:

1) We did not stress that the mouse homolog of *Csf2ra* is autosomal and have added the line: Lines 509-510: **“Of note, *Csf2ra* is autosomal in the mouse (chromosome 19).”**

2) In the discussion about CSF2RA/placenta, we have added (lines 521-522): **“Further work to elucidate the potential mechanistic role of *CSF2RA* and its protein is needed, such as in model systems of monosomy X placental development using organoids,…”**

12. In figure 8, the authors describe inflammatory changes with an irregular villus border and they link this to changes in CSF2RA expression. However, did the authors look for other signs of perturbed immune function in their data? Could other elements of the immune system be dysfunctional?

Thank you for raising this point. As shown in old Fig. 8d (new Fig. 10d), quite a large subset of genes that are lower in monosomy X placenta contribute to processes such as T-cell activation/leukocyte activation. These could have an independent effect in the monosomy X placenta, but given the fact that CSF2RA (encoding G-CSF alpha subunit) is the stand-out PAR gene, shows strong placenta/hematopoietic system specificity, and plays a role in the immune and inflammatory activation, we are convinced that CSF2RA is a key “driver”. Reduced CSF2RA function may have an effect mediated via fetal-maternal immune interactions, or other mechanisms such as host defence/infections. We feel this could be an important area to explore in the future.

We have made the following changes:

Lines 363-366 (Results): “Furthermore, pathway enrichment analysis of genes (n=266; Supplementary Data 46) that are lower in monosomy X placenta compared to control (46,XY) identified processes linked to leukocyte activation and adhesion, immune response, and cytokine signaling (Fig. 10d).” (Specifically cross-referencing the SD46 data set)

Lines 516-518 (Discussion): “.....showed enrichment of key biological processes, such as leukocyte activation, immune responses, and inflammation, involving key genes such as XCR1, interleukin 1, and interleukin family receptors.”

Line 522-523 (Discussion): “.....using organoids, and in mediating immune defence mechanisms”

13. It is also very interesting that there does not seem to be evidence of placental mosaicism, but I wonder you can be completely sure of that? Could sampling in other places of the placenta present different results? In fact the authors do not explain where and how they sampled the placentas. This bit of information should be added.

We agree that only having two samples is not absolute proof for a lack of placental mosaicism and have now undertaken further studies. We also agree that the information about placental sampling was limited.

The fetal placenta at 11-15wpc has a different structure and consistency to term placenta. The umbilical vessels are less obvious. In all situations, large amounts of placenta were available (diameter 1-2cm), so samples (30mg) were taken at different regions, and spaced far apart in the specimen.

To address the question of sample number, we have now undertaken bulk RNA-Seq and extracted data on XIST for a further 6 samples from 3 of the 45,X placentae.

Thus, we now include data on n=18 independent samples from 6 placentae: 3 placentae have 4 samples taken at geographically distinct and distanced locations; 3 placentae have 2 distinct samples (as in the original submission).

We did not find any evidence of a major 46,XX component, beyond the background maternal decidua at a similar (or even lower) level than the 46,XY placentae, which are effectively controls.

Of course, this does not absolutely mean that confined placenta mosaicism is not present, or might not occur with time during gestation. We would need an extensive study of term placentae to address that. Thus, we have made several changes to the manuscript, as well as giving considerably more detail about sampling.

We have clarified this in the methods section:

Lines 592-602 (Methods): "For the placental transcriptomic study, six monosomy 45,X placenta samples were obtained together with six age matched 46,XX control placentae and six age matched 46,XY control placentae, between 11 and 15wpc. Tissues were stored in -70°C until processing. For all 45,X, 46,XX and 46,XY placentae, the entire placental tissue (approximately 1-2 cm diameter) was visualized on ice and a sample (20-30 mg) was taken for immediate RNA/DNA extraction. For the six 45,X placentae, rapid dissection of between one (n=3) and three (n=3) further samples was performed from other distinct and distanced regions of the same placenta, for the assessment of potential 46,XX confined placental mosaicism. Thus, a total of 18 distinct 45,X samples were processed from six different 45,X placentae (n=3, two samples; n=3, four samples). For histology and IHC, independent placental tissues were stored in 10% formalin."

Figure 7b (new Fig 9b) has been updated. Originally this included the XIST data for six 45,X placental samples, and six of each controls. Because this is a stand-alone question, before looking at transcriptomic differences, we have replaced Fig 7b with a graph that contains all 18 data points. We feel this is important to have in the main manuscript.

We have updated Supplementary Fig 10b (new Supplementary Fig 12b) to show the age of the samples included in this plot.

Lines 301-303 (Results): “we initially undertook SNP array analysis using DNA derived from two to four independent areas of each placenta in the 45,X group (n=6, total n=18 areas), as well as 46,XX and 46,XY controls.”

Lines 308-310 (Results): “Taken together, these data suggest that placenta mosaicism for a 46,XX cell line is not common in 45,X placenta, at least at this stage of gestation.”

Lines 471-474 (Discussion): “...suggesting that placenta mosaicism is not likely to be a common cause of fetal rescue in TS, at least at this gestational age, although placental sampling was limited to between two to four distinct regions of each 45,X placenta.”

We have also broadened the discussion immediately after this (and in response to Reviewer 3) to say:

Lines 474-481 (Discussion): “Furthermore, because the placental material was obtained following termination in the early second trimester, we do not know whether spontaneous loss might still have occurred. Using a trophoblast-like model derived from X-monosomic human induced pluripotent stem cells, Ahern et al showed impaired secretion of placental growth factor and human chorionic gonadotropin (Ahern et al, Proc Natl Acad Sci U S A 2022). We did not observe lower expression of the genes encoding these growth factors in our datasets. Nevertheless, whether there could be a tropic growth advantage for any small mosaic population of 46,XX cells to expand with time throughout gestation remains to be seen.”

We have also added an extra limitation to the study:

Lines 543-546 (Discussion): “Finally, sampling a greater number of regions of the placenta would have provided more data to exclude significant 46,XX confined placenta mosaicism, although none was seen in the 18 samples analysed from 6 placentae in the study.”

Line 1217 Figure 7 (new Fig 9) legend: “...bulk RNA-sequencing (RNA-seq) counts in the 45,X placenta group (n=18) compared to 46,XX placenta (n=6) and 46,XY placenta (n=6).”

14. The statistical analysis seems appropriate.

We are pleased that this Reviewer, who has expertise in Turner syndrome and sex chromosome analysis, supports our approaches for analysis.

Reviewer #2 (Remarks to the Author):

In this study, Suntharalingham et al. have evaluated the transcriptional profile of various clinically-relevant tissues from late first trimester to early second trimester fetuses with monosomy X (45,X) compared to gestational-age and tissue-matched 46,XX and 46,XY controls to study differences in early fetal development associated with monosomy X and identify potential developmental origins and underlying mechanisms of the various phenotypic presentations of monosomy X/Turner syndrome.

This is a unique study and is the first to assess the transcriptome of these diverse tissues from 46,XX, 46,XY, and 45,X fetuses. Some results replicate findings that have been reported in previous studies of monosomy X from other tissues and/or stages of life, while other findings are original and add to our understanding of the monosomy X transcriptome during development – both are potentially important to provide insights into the underlying pathogenic mechanisms of this aneuploidy. These results should all be interpreted with caution however, as the small sample sizes (differential gene expression analyses based on groups of maximum N=4 for other fetal tissues and N=6 for placenta) are a limitation to drawing strong conclusions from the results of this study. This is potentially understandable given the source of the samples used for this study, and while results are novel and interesting to the field, the report could merit from being framed as an exploratory or pilot study. By making the RNA-seq data generated in this study publicly accessible, the authors have provided an excellent resource for gene expression studies in these unique and rare tissue samples for future studies. They have also provided extensive supplemental data to be transparent about the results of the study.

We are pleased that the reviewer found this study “unique” and interesting, and appreciates the importance of the data being available for the scientific community.

We also appreciate the comment about the availability of samples and that sample number are a potential limitation, and we have discussed this in the paper. In many studies using bulk RNA-seq, n=3 is the standard, especially when comparing two different tissues. We used n=4 (and n=6 for placenta) because differences were more subtle when comparing the same tissue across a narrow time frame. In fact, that is a major strength of our study design: namely, transcriptomic differences comparing different tissues or markedly different developmental stages are largely adjusted for, so the differences we see are much more likely to be related to sex chromosome effects.

Another key consideration is that differences in the fold-change of any transcript between two sample/sample sets (e.g. 45,X versus 46,XX or 45,X versus 46,XY) should be fixed, independent of the sample number. For example, if a gene has half the expression in 45,X tissue compared to control, this fold change should be the same whether just one pair of samples or ten pairs of samples are used. Of course, increasing the sample size reduces the variability, reduces the risk of chance findings and increases the power to detect smaller differences, so has an important effect when using an adjusted p value cut off. That is one reason we sought replication of consistent findings across multiple independent tissues.

We have made the following changes to put this in perspective:

Lines 36-37 (Abstract): “We therefore **undertook an exploratory study of the transcriptomic landscape.....”**

Lines 118-119 (Introduction, lead sentence): “.....weeks post conception (wpc). **Although we were limited by sample availability to some extent, our aim was to better**

understand the transcriptomic events associated with monosomy X in early human development,.....

Lines 168-169 (Results): “.....differences in gene dosage, especially related to haploinsufficiency effects in monosomy X, **and within the context of small sample sizes.**”

Lines 428-429 (Discussion): **“Further studies with larger sample numbers, and looking more specifically at developing aortic tissue, are needed.”**

The sample size is also discussed as the first point in the limitations: (Lines 529-531) **“Firstly, access to tissues of interest was limited so only a relatively small number of samples were available and within a relatively narrow developmental time window.....”**

We have also generated and analysed a combined “multi-tissue” dataset (n=20 for each karyotype) – see responses to point 1.

We would counter-argue that the consistent identification of differences in key PAR genes as well as autosomal genes such as OVCH1-AS1, shows that strong changes do emerge.

Specific comments:

1) The small sample sizes used in the various differential gene expression comparisons are a potential limitation to drawing strong conclusions in this study. With only 4 individuals per group for the fetal tissues comparisons and 6 for placental comparisons, any stochastic differences in inter-individual gene expression cannot be well controlled for and may influence results. The authors somewhat address this indirectly through the Principal Components Analysis (PCA), showing that top drivers of variation in gene expression among all tissues is tissue type, and within most individual tissues is karyotype, and also by looking for expression differences that are in common across tissues and across the 46,XX and 46,XY comparisons to try to identify replicated differences. Despite this, further evaluation of drivers of gene expression variation were not assessed, or are not mentioned in the manuscript, beyond Principal Components 1-2. Particularly within the “all tissue” PCA, were other factors such as gestational age or individual (e.g. each genetically unique individual – because multiple tissues were contributed from the same individual, particularly for 45,X cases) assessed for association with further PCs? This could help emphasize whether differences by karyotype outweigh any potential inter-individual differences, and strengthen the findings despite the small sample sizes.

As part of the PCA analysis, we routinely undertaken a Scree plot to assess the contribution of principal components in the dataset. In the “all tissues” analysis (Fig 1b), PC1 and PC2 accounted for major components related to tissue origin. Further PC3 and PC4 components were lower (10% or less), and did not show any clustering based on gestational age nor on the individual, when multiple samples came from the same individual (“all tissue” plot). We have now also analysed the data down to PC9 and do not see effects of gestational age or individual based on PC data. Indeed, some of the tissues studied (pancreas, kidney, liver) show more growth between 11-15wpc, rather than differentiation. The skin does show more differentiation and the older samples may cluster more with PC1 in this tissue (Fig. 1f). However, all tissue studies were very carefully balanced for gestational age of controls, so we do not feel that gestational age (within this relatively narrow time band) affects the main findings.

Regarding PCs for each individual tissue (e.g. Fig 1c-f), the first two PCs were in the order of PC1 25-49% variance, and PC2 15-20%. The PC values under this were lower. All samples in each tissue study were from different fetuses, and matched for gestational age and karyotype, except for the pancreas that was split. As discussed in detail in response to Reviewer 1 (point 5), the two samples from 13wpc showed biological variability and were not superimposed in the plot.

We do agree that the question of individual genetic contribution is important in our new “multi-tissue” analysis (combining different tissues, see below), and we have addressed this in response to Reviewer 3.

In addition to further support from the PCA, the authors could consider framing this study as more of an exploratory or pilot study given the small sample sizes and being more upfront about this throughout the manuscript – e.g. reminding readers of sample sizes for each of the comparisons by providing N’s within the text of results section and in the figures.

Please the comments above in relation to this Reviewer’s general comments, where we frame this as an “exploratory study” in the abstract, and stress the sample sizes and potential limitations throughout. Samples sizes are clearly stated in study design, results and figure legends. As stated above, we have made the following changes:

Lines 36-37 (Abstract): “We therefore **undertook an exploratory study** of the transcriptomic landscape.....”

Lines 118-119 (Introduction, lead sentence): “.....weeks post conception (wpc). **Although we were limited by sample availability to some extent,** our aim was to better understand the transcriptomic events associated with monosomy X in early human development,.....”

Lines 168-169 (Results): “.....differences in gene dosage, especially related to haploinsufficiency effects in monosomy X, **and within the context of small sample sizes.**”

Lines 428-429 (Discussion): “**Further studies with larger sample numbers, and looking more specifically at developing aortic tissue, are needed.**”

“MULTI-TISSUE ANALYSIS”

Although our focus was originally on looking for consistent differences between karyotypes when individual tissues were analyzed, we have now also undertaken differential gene analyses (45,X versus 46,XX; 45,X versus 46,XY) for a larger “multi-tissue group” that includes the liver, kidney, pancreas, skin and mixed group samples. Thus, there is a n=20 sample size in each of the 45,X, 46,XX and 46,XY groups (total n=60), instead of n=4. Data are now available as new datasets of 45,X versus 46,XX and 45,X versus 46,XY, Supplementary Data 38-41.

This approach increases the power to detect smaller differences in log2FC between karyotypes, and reduces tissue specific effects.

Using this approach, we did not see a great increase in the numbers of genes detected using the relatively low log2FC > 0.5 and nominal p-adj < 0.05.

The number of differentially-expressed genes in this “multi-tissue group” compared to individual groups is shown in the table below.

Differentially-expressed genes (log2FC > 0.5 and p-adj < 0.05):

Tissue	Group (n)	45,X vs 46,XX Higher mono	45,X vs 46,XX Lower mono	45,X<46,XY Higher mono	45,X>46,XY Lower mono
Pancreas	4	18	22	55	58
Liver	4	10	21	10	30
Kidney	4	22	12	10	26
Skin	4	3	8	8	60
Mixed	4	1	2	1	11
“Multi-tissue”	20	5	14	5	37

Looking at genes that were consistently higher or lower in monosomy X compared to both 46,XX AND 46,XY datasets generated the following:

Tissue	Group (n)	Higher monosomy (vs 46,XX AND 46,XY)	Lower monosomy (vs 46,XX AND 46,XY)
Pancreas	4	10	14
Liver	4	2	8
Kidney	4	6	6
Skin	4	2	4
Mixed	4	1	0
“Multi-tissue”	20	2	7

The genes identified were:

Consistently higher in monosomy X (n=2)
OVCH1-AS1, PAEP

Consistently lower in monosomy X (n=7)
AKAP17A, ASMTL, GTPBP6, PLCXD1, POLRMTP1, SLC25A6, ZBED1

We have added these data as new “Multi-tissue” Venn diagrams in Supplementary Figure 4 (Fig. 4f, lower in 45,X) and Figure 7 (Fig. 7f, higher in 45,X).

Supplementary Figure 4. Genes with lower expression in monosomy X samples. Overview of Venn diagram comparisons for all tissue groups studied: a Pancreas; b Liver; c Kidney; d Skin; e Mixed; f Multi-tissue. For each 45,X versus 46,XX tissue study group, n=4; for each 45,X versus 46,XY tissue study group, n=4. Genes included where log2 fold change <-0.5; adjusted p-value. The multi-tissue group comprised all samples in groups a to e; n=20 for each karyotype.

Supplementary Figure 7. Genes with higher expression in monosomy X samples. Overview of Venn diagram comparisons for all tissue groups studied: a Pancreas; b Liver; c Kidney; d Skin; e Mixed; f Multi-tissue. For each 45,X versus 46,XX tissue study group, n=4; for each 45,X versus 46,XY tissue study group, n=4. Genes included where log2 fold change >0.5; adjusted p-value. The multi-tissue group comprised all samples in groups a to e; n=20 for each karyotype.

We have made the following changes in the manuscript:

Lines 271-278 (Results):

Multi-tissue analysis

Given the relatively small sample size in each tissue studied (n=4 for each karyotype), an additional “multi-tissue” analysis was undertaken that combined the pancreas,

liver, kidney, skin and mixed groups (n=20 for each karyotype). Despite having more power to detect small changes, a similar group of genes was identified that were consistently lower (*AKAP17A1*, *ASMTL*, *GTPBP6*, *PLCXD1*, *POLRMTP1*, *SLC25A6*, *ZBED1*) or higher (*OVCH1-AS1*, *PAEP*) in monosomy X tissues ($\log_2FC > 0.5$, $p\text{-adj} < 0.05$) (Supplementary Fig. 4f, 7f; Supplementary Data 38-41).

Lines 534-535 (Discussion, limitations): “Furthermore, a combined “multi-tissue” analysis provided additional data.”

Lines 587-589 (Methods): “A larger “multi-tissue” group was generated (n=20 each karyotype), by combining data from these samples (pancreas, kidney, liver, skin, mixed group).”

Supplementary figures 4 and 7 have an additional panel (f, shown above), and full data are in the new Supplementary Data 38-41.

The combined “multi-tissue” data were also used to generate correlation plots, which have provided useful visualization of the range of data (see response to Reviewer 3, point 2 - new Fig. 8 and results).

2) In Lines 106-108, the authors mention how factors such as medication, inflammation, diet, etc. may confound transcriptomic profiles of adult tissues, however they do not address potential confounders of fetal or placental tissue within their own study design in the manuscript. Biological factors such as maternal health conditions or exposures can and have been shown to influence transcriptional profiles of fetal tissues including placenta (e.g. maternal cigarette smoking, maternal diabetes); other yet-detected sub-chromosomal genetic anomalies that may influence early development were not screened for and may also be possible confounders in this study. Additionally, technical factors such as sample processing time may also influence transcriptional profiles. Given that these tissues were acquired from a biobank, such information may not be readily available, but should at least be addressed as a potential confounder or limitation of the study, particularly given the small sample sizes.

We thank the reviewer for these points. We agree that other materno-fetal factors could influence fetal transcriptomics, but – because of the balanced design of the study, with carefully matched controls, any confounding effect would mostly be relevant if it affected one group more than others. As the reviewer states, there is a chance of this happening with small sample sizes, but this should have been less apparent in the multi-tissue study. Also, external drivers such as smoking – if imbalanced in one group leading to transcriptomic effects, would have lead to false positive results, and we did not see many positive differences.

To address each point, the amount of additional information available to us from the HDBR biobank about maternal smoking and health conditions is currently very limited because of the anonymization, and not sufficient for any subgroup analysis. Maternal diabetes is generally later in pregnancy, and we are not aware of diabetes in this cohort. Unlike many studies, we have performed array analysis on all samples here. The control samples were not known to have anomalies, and any samples with known anomalies were excluded of course. We cannot exclude additional (monogenic) genetic conditions by chance, but assume these would be at a level of the background population, and control samples with detected fetal anomalies were excluded.

We have made the following changes:

Lines 569 (Methods): “Control samples (46,XX, 46,XY), matched for tissue and age, were also obtained from the HDBR. **Those with any other chromosomal or other developmental anomalies were excluded.**”

We still feel that the balanced control groups and narrow time windows mean that there are likely to be fewer confounding variables compared to studying adult tissues, where women with Turner syndrome may have more treatments (e.g. blood pressure, diabetes, cholesterol lowering, anti-inflammatory medication, oestrogen replacement), compared to control men or women of the same age.

The technical issues regarding the standard operating procedures and times, sampling, and RIN values are hopefully addressed in our extensive responses to Reviewer 1, especially points 2 and 13. New data are included in the manuscript.

3) Given that a unique contribution of this study is the use of these various disease-relevant tissues, while reporting on differentially expressed genes between the 45,X and controls that are in common between each of the tissues is of interest, the authors could comment on the differentially expressed genes that may be unique within each given tissue between the Turner and control samples. Particularly if these genes are key for the function of these tissues, or are uniquely expressed in these tissues, this could be worthwhile to highlight as potential insight into the different Turner syndrome associated phenotypes observed within these tissues. This is somewhat shown in Supplementary Figures 4 and 7 but could be addressed more definitively within the manuscript.

Thank you for raising this point. We had hoped that we would see more tissue-specific effects as the reviewer suggests, but these were limited. In part, this may be due to the small sample size in each group, as the reviewer highlights, and also the relative early and narrow developmental time frame of the study.

The greatest number of differentially expressed genes between 45,X and controls was in the pancreas, but there were still only 14 genes with consistently lower expression in monosomy X. If we remove the PAR genes, we do see a group of genes that have a role in connective tissue/extra cellular matrix organization. These include *NRK* (an X chromosome gene, we have previously worked on in adrenal development), *OSR1* (a transcription factor, also in our developmental datasets), and *PDGFRA*. Furthermore, dropping the p-adj significance to 0.1 and removing PAR genes, then undertaking pathway analysis also reveals an extra cellular matrix enrichment. *NRK*, *OSR1* and *PDGFRA* may have important developmental organization effects, but none of these genes show strong pancreas specificity in the adult (Human Protein Atlas) and were not found as DEGs in the fetal kidney nor liver in our data. *SLC3A1* is differentially expressed in the kidney, pancreas and small intestine in adults, but was not found to be differentially expressed in in our kidney analysis.

On balance, we feel we do not want to speculate too much here, so have not discussed this further in the manuscript at this point.

4) In lines 275-285, the authors present their findings of a lack of mosaicism for 46,XX cells in the 45,X placentas (N=6), concluding that this mosaicism is likely not common. This is an interesting finding, and they discuss how this potentially challenges previous hypotheses that cases with 45,X that survive to term are mosaics for other viable karyotypes. However, this conclusion should be put in context of a) the small number of placental samples tested b) the timing of sample collection – given that ~98% of recognized 45,X pregnancies do not

survive to term, based on the timing of 45,X losses, is it possible that some of these fetuses may have been spontaneously lost later in pregnancy and represent part of this larger group of pregnancies that are not “rescued”?; and c) any potential ascertainment bias – because samples included in this study were selected on the basis of having a 45,X karyotype, could this group of placentas simply just be enriched for cases that are non-mosaic/very low-level 45,X karyotypes? Further explanation of these factors would be helpful to support the conclusion being drawn.

Thank you for your interest and for these comments, which are similar to some raised by Reviewer 1 (point 13). To address these:

a) We have increased the 45,X placenta mosaicism analysis to n=18 samples from six placenta. Three placentae were sampled twice and three were sampled in four distinct locations. No evidence of 46,XX mosaicism was found. We are also aware that there are still limitations. These points, as well as extensive information about placental sampling, have now been added, and are detailed in the response to Reviewer 1, Point 13 (changes in text, figures, additional data).

b) The very important point about potential pregnancy loss (or not) was raised by Reviewer 1 (point 4), and have been addressed in that response. Of note, we have expanded the discussion to say:

Lines 469-481 (Discussion): “Here, we did not identify significant 46,XX mosaicism using both RNA (*XIST* counts) and DNA (SNP arrays) technologies in the six 45,X placental samples studied, suggesting that placenta mosaicism is not likely to be a common cause of fetal rescue in TS, at least at this gestational age, although placental sampling was limited to between two to four distinct regions of each 45,X placenta. Furthermore, because the placental material was obtained following termination in the early second trimester, we do not know whether spontaneous loss might still have occurred. Using a trophoblast-like model derived from X-monosomic human induced pluripotent stem cells, Ahern et al showed impaired secretion of placental growth factor and human chorionic gonadotropin. We did not observe lower expression of the genes encoding these growth factors in our datasets. Nevertheless, whether there could be a tropic growth advantage for any small mosaic population of 46,XX cells to expand with time throughout gestation remains to be seen.”

c) The placenta samples were enriched for 45,X karyotypes, as opposed to 45,X/46,XX mosaicism or other X variants. But we know that approximately 50% of women with Turner syndrome have a 45,X karyotype (on blood), and extensive mosaicism is not common when reassessed in adulthood (e.g., Suntharalingham et al, 2023 PMID: 37800145). So even if we focused on a 45,X status here, we feel it is representative of a population that do survive to term.

5) In line 339, the authors report on results of pathway enrichment for genes with lower expression in 45,X placenta vs 46,XY placenta, highlighting enrichment of processes related to immune function. They do not however comment on any pathway enrichment in 45,X vs 46,XX placentas. Was this performed? If so, this should be mentioned and a comment on whether the immune-related pathway enrichment is consistent or not. If it was not performed, a justification for this should be given.

We focussed on the 45,X vs 46,XY placenta comparison as this provided a sufficient number of differentially expressed genes with the significance cut-off to undertake a meaningful pathway analysis (n=266). The number of differentially expressed genes in

the 45,X vs 46,XX analysis was not enough. These data are available in supplementary data. We have also changed the text to read:

Lines 1260-1261 (New Fig. 10 legend): **“The number of differentially expressed genes in 45,X placenta compared to 46,XX controls was insufficient for pathway analysis.”**

6) While not a study of direct fetal tissues such as this one, a pilot study by Massingham et al. (2014) also assessed transcriptional differences in 45,X vs 46,XX fetuses based on cell-free RNA in amniotic fluid (AF). Cell-free nucleic acids in AF are primarily fetal in origin and expected to arise from fetal tissues in direct contact with AF such as skin, therefore given that this is one of the few other studies assessing transcriptional differences in Turner syndrome in prenatal development, a mention and comparison to the results of this study would be a helpful addition to the discussion. In particular, the authors have some similar findings as in this study, eg. SHOX not differentially expressed, low expression of LDLR.

Massingham, L. J., Johnson, K. L., Scholl, T. M., Slonim, D. K., Wick, H. C., & Bianchi, D. W. (2014). Amniotic fluid RNA gene expression profiling provides insights into the phenotype of Turner syndrome. *Human genetics*, 133(9), 1075–1082.

We thank the reviewer for highlighting this paper. We did not appreciate it included findings such as LDLR. Our data suggest that LDLR is lower in monosomy X. This finding would be consistent with a profile of elevated cholesterol and potential risk of atherosclerosis. The Massingham et al study (2014) on amniotic fluid RNA gene expression (using microarrays) found a higher expression of LDLR in the samples studied following (8.15 fold, specific p-adj value not given).

Massingham et al: *“The cause of hyperlipidemia in individuals with Turner syndrome is not currently known, but the increased expression of LDLR may provide a clue to the dysregulation. Hyperlipidemia and atherosclerosis are well known to be associated with Turner syndrome”*

Cell-free RNA in amniotic fluid may also reflect kidney/bladder transitional epithelium, gastrointestinal cells and many other cell types. We have now referenced this study and provided a comment on this, although we are cautious about overplaying findings, especially as our findings of potentially lower LDLR expression make more sense biologically.

We have made the following changes:

Lines 387-389 (Discussion): **“Not all PAR genes showed lower expression, mostly due to lower background transcript levels or greater variability in expression between different tissues (e.g., SHOX). A lack of SHOX differential expression has been found previously (Massingham et al 2014).”**

Lines 416-420 (Discussion): **“For example, lower expression of the low-density lipoprotein receptor gene (*LDLR*), which regulates cholesterol homeostasis, could contribute to the risk of dyslipidemia in Turner syndrome⁴⁴. Previously, increased *LDLR* expression has been reported in amniotic fluid RNA in a small number of monosomy X fetuses (Massingham et al 2014).”**

Reviewer #3 (Remarks to the Author):

Summary & overall impression

This interesting work profiles the transcriptomes of 45,XO prenatal tissue samples using mRNA sequencing to probe the biological consequences of having only one X early in developmental in light of the high rate of spontaneous abortion seen in monosomy X conceptuses and specifically evaluates the hypotheses of reduced PAR/escape gene expression driving monosomy X phenotypes. This study is important to publish as it provides information on a timepoint that is difficult to study, particularly using a multi-tissue approach, however, the analytical methods would benefit from revision as the methods implemented are not the most statistically powerful for this study design.

We are pleased the reviewer found this study interesting and important to publish. Our responses to comments about statistical analysis are addressed below.

Comments

- Major 1: I am not convinced that the statistical approach taken (linear modelling) is valid for this study design. Due to the sample distribution presented in Supp figure 1, when comparing between tissue types the authors I think have a mixture of some genetically identical samples (with same donor contributing multiple tissues), alongside samples that are only donors for one tissue type. In this case, a linear mixed effects model controlling for individual as a random effect would be a more appropriate choice for all of the linear models presented. In addition, rather than stratifying comparisons between 45,X and the 46,XX or 46,XY samples by tissue, it would also be more statistically powerful to run one model including all samples + tissue types, with an interaction term for tissue*karyotype. This is under-appreciated, but stratifying data by sex, etc, reduces power to discover true positives. At the very least, a mixed effects analysis could be included as a supplementary sensitivity analysis. For a further discussion of this topic, see Chin & Christians 2015, PMID26283072, <https://pubmed.ncbi.nlm.nih.gov/26283072/>

Thank you for these comments. We chose to undertake an individual analysis of each tissue type with balanced groups for karyotype and gestational age, and then to identify consistent differentially expressed genes, as we felt this was the most robust approach and one that is often used. We are aware that this is a very stringent approach, but as Reviewer two points out, the sample sizes are small and we did not want to generate false positive findings. However, we appreciate this reviewer's comments (and the point about correlations below), and have put significant effort into reanalyzing data.

First, we apologise that the Supplementary Fig 1 overview of the study was potentially misleading as not all samples were included (for placenta) and the grouping of control samples was unclear. We have changed this figure to remove the placenta (which is a separate study, with larger numbers) and to clarify sample origin better. We have also included sample identifiers in Supplementary Data 1. The replacement Supplementary Fig. 1 (now Fig. 1a) is shown below. We hope this is clearer, as we now combine samples in a multi-tissue analysis, but also address the potential effect of "donor" (see below).

Tissue	11 wpc	12 wpc	13 wpc	14 wpc	15 wpc
Pancreas					
Liver					
Kidney					
Skin					
Brain (Mixed)					
Heart (Mixed)					
Lung (Mixed)					
Spleen (Mixed)					

Linear mixed effects modelling

The second point relates to linear modelling versus linear mixed effects modelling, and the potential effects of several samples being obtained from the same “donor”. We initially did not feel this would have a major effect as each tissue study was a balanced, separate analysis. The pancreas study is the only one with two samples coming from the same origin, and this is discussed in relation to the other Reviewers’ comment. Also, DESeq2 uses a binomial approach and controlling for random effects is harder. We felt that individual effects in the transcriptome are likely to be small in relation to the tissue of origin and karyotype.

Nevertheless, we acknowledge this is an important point, especially as we have now undertaken a “Multi-tissue” analysis that combines all 5 tissue groups (pancreas, liver, kidney, skin, mixed), so that there are 20 samples in each of the three groups (45,X; 46,XX; 46,XY). This is described in detail in response to Reviewer 2, Point 1). This larger “multi-tissue” analysis has more power to detect significant changes, although the final analysis was not really very different. However, this approach does involve several samples from the same “donor” so we wanted to apply a linear mixed effects approach. To do this, we had to reanalyse the dataset in limma-voom, rather than DESeq2. We then compared the results for 45,X versus 46,XX and 45,X versus 46,XY using the original DESeq2 (linear) pipeline, limma-voom non-adjusted and limma-voom adjusting for individual as a random effect (linear mixed effects). All significant results (adjusted $p < 0.05$) (irrespective of \log_2 fold change) for all the analyses are shown in New Supplementary Data file SD42 and below. Using the linear mixed effects approach to identify genes that were consistently lower in monosomy (in 45,X versus 46,XX and 45,X versus 46,XY) or higher in monosomy compared to controls, did not provide any major changes in findings. We did exclude some potential false positives with this approach, but they were excluded using our overlapping data strategy anyway.

We have made the following changes in the text:

Lines 281-284 (Results): **“As the multi-tissue analysis contains several different tissue samples from the same fetus (Supplementary Fig. 1a), further analysis was undertaken using a linear mixed effects model with individual as a random effect, but no major differences were seen (Supplementary Data 42).”**

Lines 648-650 (Methods): **“A similar multi-tissue linear analysis was undertaken (n=20 each karyotype) in DESeq2, as well as using a linear mixed effects model in limma-voom, with individual as a random effect.”**

Interactive terms

We appreciate the comment about using interactive terms as a more powerful way to detect differences. We have therefore reanalysed the data using an interactive term for tissue and karyotype, but found slightly fewer genes (based on adj. p-value) (three 45,X vs 46,XX; four 45,X vs 46,XY) rather than detecting more potential true positives.

- Major 2: Genes with lower expression in monosomy X (line 159) and genes with higher expression in monosomy X across tissues (line 220) – intersecting adjusted p-value lists of differentially expressed genes is not a sufficiently powerful way to conduct this type of comparative analysis and leads to overestimation of false negatives or lack of overlap (see reference suggested above, Chin & Christians 2015). A stronger alternative would be to consider whether the effect sizes of DEGs at nominal significance are correlated between tissues (i.e. logFC correlations between tissue types) and to select those that are most strongly correlated to determine a consensus list of genes that are affected by karyotype.

We agree that intersecting results for individual studies could generate false negatives, but we wanted to adopt a stringent approach. We have therefore also undertaken a larger “multi-tissue” analysis as discussed above (and Reviewer 2), with larger sample sizes to generate a better core group of genes. This is discussed as a new subsection of the results, and in relation to linear mixed effects above.

In addition, we have taken on board the reviewer’s comments about correlation analysis. We did feel we could convey a large amount of source data succinctly by using “jitter plots” with colour-coded defined data points for all five tissue analyses (liver, pancreas, kidney, skin, mixed) superimposed on the mean value (e.g. Figs. 3e, 4a&b, etc). However, as suggested, we have now undertaken plots of nominally significant genes in all tissues, and present the comparative plots of these for all six tissue permutations in new Supplementary Figure 9 (for 45,X versus 46,XX) and in new Supplementary Figure 10 (for 45,X versus 46,XY). We included genes with nominal significance. Correlation coefficients are provided, and key genes are labelled. These data clearly show consistently higher expression of *OVCH1-AS1*, and lower expression of the genes identified previously (e.g. *PAR1* genes).

Supplementary Figure 10. Correlations of log₂FC differences for 45,X vs 46,XX between tissue groups. a pancreas vs liver; b liver vs kidney; c pancreas vs kidney; d liver vs skin; e pancreas vs skin; f kidney vs skin. Genes were included where p<0.05.

Supplementary Figure 11. Correlations of \log_2FC differences for 45,X vs 46,XY between tissue groups. a pancreas vs liver; b liver vs kidney; c pancreas vs kidney; d liver vs skin; e pancreas vs skin; f kidney vs skin. Genes were included where $p < 0.05$.

We also generated a correlation plot for log₂FC of 45,X versus 46,XX (y-axis) against log₂FC of 45,X versus 46,XY (x-axis) of all gene transcripts for all individual tissue analysis (ie approximately 30,000 gene transcripts each represented five times: pancreas, kidney, liver, skin, mixed). Data are shown in new Figure 8 and below.

Figure 8. Correlation plot of gene expression in 45,X tissues compared to 46,XX (y-axis) and 46,XY (x-axis) showing global patterns of differential changes. Each data point represents a specific gene in each of five different tissue groups (pancreas, liver, kidney, skin, mixed group). Genes with higher expression in monosomy X are shown in the upper, right quadrant (e.g., *OVCH1-AS1*). Genes with lower expression in monosomy X are shown in the lower, right quadrant. Key clusters of Y chromosome genes and X-inactivation genes (e.g. *XIST*) are also shown.

We feel this is a very useful way of presenting the whole data-set in a single graphical representation, as it shows both the number and magnitude of differentially expressed genes in both the 45,X versus 46,XX and 45,X versus 46,XY datasets. This approach clearly shows Y-specific genes (left on the x-axis), 46,XX specific genes (*XIST*, *TSIX*) (low on the y-axis), and the correlation of genes in both datasets that are lower in 45,X (lower left) and higher in monosomy X (upper right). As the labels for individual genes are in the correlation plots, we felt this graphic was more useful to show the overall distribution of data and to support the basic concepts above

We have changed the text to:

Lines 278-281 (Results): “By correlating the log₂FC for 45,X versus 46,XX with the same gene in 45,X versus 46,XY datasets, the distribution and magnitude of differentially expressed gene could be clearly seen (Fig. 8; see Supplementary Figs. 10 and 11 for individual tissue correlation plots).”

3) - Sampling methods of the clonally-developed placenta should be described as thoroughly as possible in the Methods section, though I understand the difficulty associated with early-gestation placental sampling.

Thank you for this important point. We have now included much more detail about sampling methods for the placenta, as detailed in the response to Reviewer 1 (point 13) and Reviewer 2 (point 4). The placenta at this age is different in structure and consistency to term placenta. We have also increased the analysis to look at four regions from three placentae (and two from the other three), meaning that a total of 18 samples were studied from six different placentae. All our responses and changes to the text are outlined above, in response to Reviewers 1 and 2.

4) - What were the results of RNA sample QC with TapeStation prior to sequencing (Methods line 546)? It is important to report the metrics assessed (RIN, DV200, etc) as well as the average results of these assessments, as I imagine tissue may be confounded with RNA quality, and that context is important for manuscript interpretation.

Thank you for raising this point, as requested by other reviewers (especially Reviewer 1, point 2). The key RIN values for all different karyotypes in the main tissue groups are now presented in new Supplementary Figure 1. They are all good quality and with no significant differences between karyotype groups. The placenta RIN values were lower, as is often the situation due to high RNase activity, but adequate libraries were generated from these samples.

We also have added substantially more detail about tissue collection, transport times and other variables, where available, as detailed in the responses to Reviewers 1 & 2.

5) As noted by the authors, in the introduction (line 87, reference 25), San Roman et al. recently identified X-linked genes that were proposed to drive somatic sex differences, the relative expression of the genes ID'd by San Roman are also evaluated explicitly in the Results section and reported on in Supp Data 30. However, more recent work has been published by the same group defining autosomal genes that are moderated by the impact of an inactive X or Y chromosome. The loss of either of these would lead to monosomy X and it could be presumed for similar reasons that many of these genes may be affected in 45,XO samples. As such it would be valuable to extend the analysis in lines 181-192 to include the most recent San Roman et al. 2024 findings and evaluate the effect of karyotype on genes, or use this list to contextualize the findings presented in lines 194-212. San Roman et al. 2024: PMID38190107 <https://www.ncbi.nlm.nih.gov/pmc/articles/PMC10794785/>

Thank you for this suggestion to further analyse autosomal gene expression in relation to this recent paper (San Roman et al. The human Y and inactive X chromosomes similarly modulate autosomal gene expression. Cell Genom 2024 4(1):100462).

To try to relate these findings to our datasets, we have looked at available supplementary data (Supplementary Data/Table 2), which focusses on potential effects of X chromosome gene dosage in lymphoblastoid cell lines (LCL) and fibroblasts, including from multiple individuals with 45,X and 47,XXY karyotypes. Because of the large sample size (n>150), a great number of autosomal genes were found that might be altered by X chromosome number, but the magnitude of this effect was often small. It seems that genes with low transcripts were excluded, and the authors show in general somewhat limited overlap between these cell types (see below). Importantly in the fibroblast dataset there are a large number of genes with a

“NA” designation, including genes such as *OVCH1-AS1*, so we decided to analyse the LCL and fibroblast data independently rather than looking at just shared genes.

We used various cut offs in our autosomal datasets. We used the mean data between tissues (cut off FC0.35 irrespective of significance) and our consistent finding was *OVCH1-AS1*. We also used our new multi-tissue analysis, with more power to detect consistent effects. Here, we considered all significant lower and higher differentially expressed genes, in the 45,X versus 46,XX and the 45,X versus 46,XY datasets, irrespective of fold change.

The Venn diagram for overlap is shown below.

Unfortunately, we did not identify any key autosomal genes or pathways that were differentially expressed in our data and in San Roman 2024, except for *OVCH1-AS1*, which once again was the stand out gene. It was not found in the San Roman fibroblast data as it was “NA” (Table S2). We did see a much greater overlap in our 45,X versus 46,XX and 45,X versus 46,XY multi-tissue datasets (n=21, approximately 40% of genes), compared to the LCL vs fibroblast datasets (approximately 15-25% of genes), but again some of the fold changes we observed are very low or not consistent, and the 45,X dataset is a common comparator, so we do not want to overinterpret these data at present.

Since submitting our manuscript, we have also seen a bioRxiv manuscript from the Page group, that looks at similar effects and autosomal gene regulation in four cells (CD4+cells, monocytes, lymphoblastoid cell lines (LCLs) and fibroblasts) across a range of karyotypes (1-3 X chromosomes, 0-2 Y chromosomes) (doi: 10.1101/2024.03.18.585578). This work is now published (Blanton et al. Stable and robust Xi and Y transcriptomes drive cell-type-specific autosomal and Xa responses

in vivo and in vitro in four human cell types. Cell Genom 2024 4(9):100628). This paper suggests that autosomal gene effects of X chromosome gene dosage may be cell-specific. This may be an important concept but one we cannot address easily at present with our sample size.

6) - Supplementary Figure 1 is slightly confusing, in addition to being biological replicates if two half circles are present in the same tissue type, they also appear to be technical replicates? I.e. do two half-circles (purple) in the pancreas category indicate that the same pancreas sample was transcriptome sequenced 2x? If so, were these technical replicates removed prior to linear modelling? Were they used for any other quality assessments? This should be clarified and confirmed in the results section of the manuscript around line 149.

As noted above, we agree with the Reviewer that Supplementary Figure 1 is confusing. Also, we did not include all control placental samples so it was not accurate. We have modified it as shown in response to this Reviewer's Major point 1 (above).

The issue about the pancreas sample is discussed in detail in response to the other Reviewers. We feel it is not a technical replicate (i.e., using the same extracted RNA twice), but more a biological replicate, especially given the variant on PCA (see Reviewer 1 point 5). Bisecting a structure at this age results in variability in tissue components. We have edited the results and legends accordingly, as described above (e.g. in relation to Reviewer 1, Point 5).

7) - Additionally, was gestational age (or any other covariate) accounted for in the linear model presented in Figure 2?

We did not account for gestational age in Fig 2 as there samples were all from a reasonably narrow developmental time window, and the study was designed so that 45,X samples were all tightly matched with controls for age. We have also discussed this in response to Reviewer 2, points 1 and 2. We have looked at different principles components and this is not a clear issue.

8) - I appreciate that placenta was not included in the DEG analyses presented in Figures 2-5 as it is analysed separately in the later section, however the manuscript would benefit from addressing the motivation for analyzing placenta separately earlier, i.e., in the introduction. Additionally, a potentially useful paper to add to your evidence for placental intolerance of monosomy X would be Ahern et al. 2022: <https://pubmed.ncbi.nlm.nih.gov/36161909/>

We focussed on the placenta separately, in order to address the mosaicism question and to investigate potential molecular mechanisms for pregnancy loss. On re-reading the introduction, we agree that the placental component of the study is not well defined. We have therefore clarified the lead sentence of the introduction to say:

Line 121-122 (Introduction, Lead sentence): ".....to obtain a unique perspective on human X chromosome biology and possible disease mechanisms in Turner syndrome, and to investigate the effects of monosomy X in the placenta in relation to pregnancy loss."

We should have included the Ahern paper (albeit a model system) and have added this to a much broader discussion to say:

Lines 469-481 (Discussion): “Here, we did not identify significant 46,XX mosaicism using both RNA (*XIST* counts) and DNA (SNP arrays) technologies in the six 45,X placental samples studied, suggesting that placenta mosaicism is not likely to be a common cause of fetal rescue in TS, at least at this gestational age, although placental sampling was limited to between two to four distinct regions of each 45,X placenta. Furthermore, because the placental material was obtained following termination in the early second trimester, we do not know whether spontaneous loss might still have occurred. Using a trophoblast-like model derived from X-monosomic human induced pluripotent stem cells, Ahern et al showed impaired secretion of placental growth factor and human chorionic gonadotropin (Ahern et al., 2022). We did not observe lower expression of the genes encoding these growth factors in our datasets. Nevertheless, whether there could be a tropic growth advantage for any small mosaic population of 46,XX cells to expand with time throughout gestation remains to be seen.”

Another reason for considering the placenta as a separate standalone analysis, is that we were not sure whether we would have substantial components of maternal decidua present (46,XX), which might have complicated the analysis if we merged this sub-study with the other tissue datasets.

9)- It may be more valuable to present supplementary Figure 5 sorted by karyotype on the X axis and colored by tissue.

We have considered this point, but prefer to keep it as it is as we are looking at comparisons between karyotype within tissues. Analyzing between tissues make individual gene comparisons less obvious and harder to interpret (especially as it is a log2 scale and differences are less obvious visually).

10) - The logic for only analysing chromosome 12 genes related to the lncRNA expression was not clear in the paper, I assume it was simply due to a higher probability of detecting cis effects? This could be clarified.

As *OVCH1-AS1* is a chromosome 12 we undertook this whole chromosome plot to highlight that *OVCH1-AS1* is the stand out gene, to confirm the violin plots and independent qRT-PCR that genes in the region of *OVCH1-AS1* did not show differential regulation, and that there were no long-range cis-regulatory effects. We have clarified this by saying:

Lines 454-457 (Discussion): “As antisense lncRNAs sometimes modulate sense strand RNA, or genes in the region, we undertook more detailed analysis of the *OVCH1-AS1* locus of chromosome 12 to investigate potential cis effects, but we did not see an obvious effect on the expression of *OVCH1* nor of other local genes.”

We have also looked at interesting Chromosome 12 (autosomal) genes in other datasets (e.g. San Roman 2024, as suggested by this Reviewer). Although altered regulation of several chromosome 12 genes has been suggested, there did not appear to be an enrichment of chromosome 12 genes and we could not find any genes in proximity to *OVCH1-AS1*.

ADDITIONAL CHANGES BY THE AUTHORS

1) We have cited the new Turner Syndrome Clinical Guidelines in the introduction

Line 55. Gravholt CH, Andersen NH, Christin-Maitre S, Davis SM, Duijnhouwer A, Gawlik A, Maciel-Guerra AT, Gutmark-Little I, Fleischer K, Hong D, Klein KO, Prakash SK, Shankar RK, Sandberg DE, Sas TCJ, Skakkebaek A, Stochholm K, van der Velden JA; International Turner Syndrome Consensus Group; Bacheljauw PF. Clinical practice guidelines for the care of girls and women with Turner syndrome. *Eur J Endocrinol.* 2024 Jun 5;190(6):G53-G151. doi: 10.1093/ejendo/lvae050. PMID: 38748847

2) After discussion with several parties, including support groups, we are keen to put the issue of pregnancy loss in a clearer context, and avoid numbers based on limited data. We have therefore modified the introduction to:

Lines 55-59: “Although monosomy X is the only chromosome monosomy compatible with survival in humans, it is estimated that only a proportion of monosomy X fetuses survive to term and many pregnancies are spontaneously lost in the first or early second trimester, often before a pregnancy is recognized⁵⁻⁷.”

Line 464: “....It has been reported that many monosomy pregnancies are lost mostly towards the end of the first trimester.....”

3) We have updated references to include recent PageGroup data (as analyzed above in response to Reviewer 3)

San Roman AK, Skaletsky H, Godfrey AK, Bokil NV, Teitz L, Singh I, Blanton LV, Bellott DW, Pyntikova T, Lange J, Koutseva N, Hughes JF, Brown L, Phou S, Buscetta A, Kruszka P, Banks N, Dutra A, Pak E, Lasutschinkow PC, Keen C, Davis SM, Lin AE, Tartaglia NR, Samango-Sprouse C, Muenke M, Page DC. The human Y and inactive X chromosomes similarly modulate autosomal gene expression. *Cell Genom.* 2024 Jan 10;4(1):100462. doi: 10.1016/j.xgen.2023.100462. Epub 2023 Dec 13. PMID: 38190107

Blanton LV, San Roman AK, Wood G, Buscetta A, Banks N, Skaletsky H, Godfrey AK, Pham TT, Hughes JF, Brown LG, Kruszka P, Lin AE, Kastner DL, Muenke M, Page DC. Stable and robust Xi and Y transcriptomes drive cell-type-specific autosomal and Xa responses in vivo and in vitro in four human cell types. *Cell Genom.* 2024 Sep 11;4(9):100628. doi: 10.1016/j.xgen.2024.100628. Epub 2024 Aug 6. PMID: 39111319

4) We have changed the “Statistical analysis” section of the methods to a “Statistics and reproducibility” section, as per guidelines (lines 727-741). We have added data about sample number, replicates and analysis, although this is covered in the methods, results, figure legends and supplementary data.

RESPONSES TO REVIEWERS

COMMSBIO-24-0843-T

The transcriptomic landscape of monosomy X (45,X) during early human fetal and placental development

Suntharalingham et al.

Reviewer #1 (Remarks to the Author):

No additional comments. The authors have done a very good job in responding to my comments.

Thank you.

Reviewer #2 (Remarks to the Author):

I thank the authors for their detailed responses and modifications to the original manuscript to address my and other reviewers' comments. The additional details and analyses included in response to reviewer comments improve clarity and strengthen the manuscript. Based on the revised manuscript and responses provided, I believe the authors have sufficiently addressed my concerns.

Thank you.

Reviewer #3 (Remarks to the Author):

Thank you for revising your work according to our feedback. In particular,
- Thank you for including the linear mixed effects model and results. Although the results are largely consistent with your previous analysis in this dataset, it is valuable and adds confidence in your other results to see the mixed effects model, given that it is the most appropriate test for this type of analysis.

Thank you

- Fewer significant results after running an interaction test are expected, as likely you are losing the false positives. I would have appreciated seeing this addressed in at least one sentence in the manuscript.

We have included the following sentence in the results (lines 282-283):

"Fewer significant results were found due to a reduction in false positives, as expected for an interaction test (Supplementary Data 42)."

- The correlation plots largely show what the authors had earlier hypothesized, and the differences in strength of correlation in the various tissue comparisons is interesting. Clearly some tissues (as expected) are more consistent with each other than with others.

We felt these are useful additional data.